# Proinflammatory Risk Factors in Patients with Ischemic Stroke: A Systematic Review and Meta-Analysis

**DOI:** 10.3390/antiox14101229

**Published:** 2025-10-14

**Authors:** Alexandru Gerdanovics, Ioana Cristina Stănescu, Camelia Manuela Mîrza, Gabriela Bombonica Dogaru, Cristina Ariadna Nicula, Paul-Mihai Boarescu, Cezara-Andreea Gerdanovics, Adriana Elena Bulboacă

**Affiliations:** 1Faculty of Medicine, “Iuliu Hațieganu” University of Medicine and Pharmacy Cluj-Napoca, Victor Babeş Street, No. 8, 400347 Cluj-Napoca, Romania; 2Department of Neurology, “Iuliu Hațieganu” University of Medicine and Pharmacy Cluj-Napoca, Victor Babeş Street, No. 43, 400012 Cluj-Napoca, Romania; 3Department of Pathophysiology, “Iuliu Haţieganu” University of Medicine and Pharmacy, Victor Babeş Street, No. 2-4, 400012 Cluj-Napoca, Romania; 4Department of Biomedical Sciences, Faculty of Medicine and Biological Sciences, “Stefan cel Mare” University of Suceava, Univeristății Street, No. 13, 720229 Suceava, Romania

**Keywords:** stroke, ischemic, neuroinflammation, risk, factors

## Abstract

Ischemic stroke is a leading cause of disability worldwide, often triggered by atherothrombotic or embolic events. A growing body of evidence highlights the role of neuroinflammation as a central mechanism in post-stroke damage, influenced by modifiable systemic risk factors. Emerging evidence suggests that oxidative stress mediates the impact of several modifiable risk factors by activating redox-sensitive pathways (such as NF-κB), impairing nitric oxide bioavailability, and promoting matrix metalloproteinase activity that disrupts vascular integrity and contributes to ischemic injury. In this context, our meta-analysis examined major modifiable risk factors for ischemic stroke, with a particular focus on their shared ability to promote oxidative stress and neuroinflammatory cascades. By emphasizing these redox-dependent mechanisms, our work supports the biological plausibility of exploring antioxidant strategies as complementary approaches to mitigate stroke risk. Hypertension, diabetes, dyslipidemia, smoking, atrial fibrillation, and transient ischemic attacks all contribute to oxidative damage through mechanisms such as endothelial dysfunction, vascular inflammation, and excessive free radical exposure. We searched PubMed, PubMed Central, Web of Science, and Scopus for observational studies published within the last five years, identifying 23 studies (691,524 participants) meeting eligibility criteria. Using a random-effects model, we found significant associations between stroke risk and hypertension (OR = 1.58, 95% CI: 1.28–1.94), smoking (OR = 1.61, 95% CI: 1.13–2.28), type 2 diabetes (OR = 1.53, 95% CI: 1.29–1.81), atrial fibrillation (OR = 1.88, 95% CI: 1.28–2.75), and prior transient ischemic attack (OR = 1.62, 95% CI: 1.24–2.11). These risk factors are known to contribute to systemic inflammation, potentially exacerbating neuroinflammatory cascades post-stroke. Despite limitations such as heterogeneity and low certainty of evidence, our findings reinforce the relevance of targeting inflammation-driven risk factors in stroke prevention strategies and future research.

## 1. Introduction

Stroke is considered the second leading cause of death and third leading cause of disability worldwide [1]. Ischemic stroke, together with acute myocardial infarction, accounts for 85% of all cardiovascular mortality [2]. In Europe, prevalence is particularly high in Eastern countries [3], where healthcare systems face increased demands for acute care and rehabilitation. In Romania, stroke accounts for approximately 252.8 deaths per 100,000 individuals [4], with elderly patients being most affected [5]. Local analyses highlight substantial direct medical costs and gaps in neurorehabilitation services [6,7], underlining the need for targeted prevention strategies such as hypertension screening and smoking cessation.

Ischemic strokes are classified by the TOAST system into large-vessel atherosclerosis, cardioembolism, small-vessel occlusion, other determined causes, and undetermined etiology [8]. Atherothrombotic and embolic mechanisms are the most common [8]. Atherogenesis involves endothelial injury promoted by hypertension, dyslipidemia, smoking, and diabetes, leading to lipid accumulation, inflammation, and plaque rupture [9,10,11]. Ischemia initiates cascades of excitotoxicity, inflammation, and oxidative stress [12,13,14,15,16], with neuroinflammation emerging as a key determinant of neuronal survival.

Oxidative stress is a central unifying pathway in stroke pathogenesis. Defined as an imbalance between reactive oxygen/nitrogen species and antioxidant defenses [17,18,19], it promotes vascular dysfunction by reducing nitric oxide bioavailability, increasing arterial stiffness [19,20], activating NF-κB [20,21], and stimulating matrix metalloproteinases that disrupt the blood–brain barrier [21,22]. These mechanisms align with the concept of “inflammaging,” in which aging sustains chronic low-grade inflammation [23,24,25]. Thus, while stroke risk factors differ in nature—modifiable (e.g., hypertension, diabetes, dyslipidemia, smoking, alcohol, obesity, sedentary lifestyle, sleep apnea, periodontal disease, atrial fibrillation, diet) or non-modifiable (age, sex, race, genetics) [8]—they converge on oxidative and inflammatory pathways.

Several meta-analyses have assessed stroke risk across populations: global comparisons [25], ethnicity-specific analyses [26,27], recurrence studies [28,29], and sex-based differences [30]. Recent societal shifts, including lifestyle transitions and the COVID-19 pandemic, further altered risk factor prevalence [31,32]. Against this background, our study aimed to provide an updated synthesis of the most prevalent modifiable risk factors for ischemic stroke, restricted to studies published in the last five years, and to highlight their interplay with oxidative and inflammatory mechanisms. By emphasizing how these risk factors converge on redox imbalance, we also underline the potential role of antioxidant-based strategies in reducing the burden of ischemic stroke.

## 2. Materials and Methods

We conducted a systematic review following the Preferred Reporting Items for Systematic Reviews and Meta-Analyses (PRISMA) 2020 guidelines [33] of studies indexed in PubMed, PubMed Central, Scopus, and Web of Science (WOS) published within the last 5 years, focusing on risk factors in adult ischemic stroke patients.

Specific inclusion and exclusion criteria were applied in this study. We included different types of studies (observational, cohort, case–control) focusing on adults with ischemic stroke and predefined risk factors. Studies varied in their primary objectives (incident ischemic stroke risk in general populations, recurrence risk, or outcomes in post-stroke cohorts). We included all these under the rationale that vascular risk factors exert consistent influence across the ischemic stroke continuum. However, for transparency, we labeled studies in Table 1 according to their primary outcome category (incident, recurrent, or post-stroke outcome). Studies involving only certain types of stroke, unknown causes of stroke, genetic risk factors, different treatment strategies or pediatric populations were excluded. Genetic studies were excluded as our primary objective was to evaluate modifiable, clinically actionable risk factors. While genetic predispositions are important in stroke research, they do not directly translate into immediate prevention strategies, which was the focus of our review. One GRS study is retained descriptively for context and not pooled. Similarly, non-modifiable determinants such as aging and sex, as well as poorly standardized or inconsistently reported contributors such as viral or bacterial infections, were not included in the quantitative synthesis. These factors, although biologically relevant, lacked sufficient homogeneity across the recent literature to allow robust meta-analytic assessment. Our focus was therefore narrowed to modifiable, clinically actionable risk factors, aligning with the primary scope of informing prevention and rehabilitation strategies.

Our comprehensive search strategy included several databases. PubMed and PubMed Central were the first to be accessed in March 2025, then we extended the search with Scopus and Web of Science during April 2025. The search strategies included Medical Subject Headings (MeSH) terms and relevant keywords. In PubMed and PubMed Central, the following MeSH-based query was used: ((“Stroke” [Mesh]) AND (“Ischemia” [Mesh]) AND (“Risk Factors” [Mesh])), on 30 March 2025. To refine our selection, we applied the following filters: Publication Date: Last 5 years, Text availability: Free full text, Language: English, Species: Humans, Age: Adults aged 19 and above.

In the Web of Science Core Collection, we performed an advanced search using the following query: (ALL = (ischemic stroke) AND ALL = (risk factors)), on 22 April 2025. To refine the search results and ensure relevance to our research objectives, several filters were applied. Specifically, the search was limited to articles published between 2020 and 2025, written in English, and available as open-access publications. In addition, we restricted the selection to original research articles, excluding reviews and retracted publications, and focused on studies classified under the research area “Clinical Neurology” with the citation topic “Strokes.”

The search strategy was also adapted for Scopus, using the following search query: “ischemic AND stroke AND risk AND factors”. The search was performed on 25 April 2025. Several filters were applied to refine the results: studies published between 2020 and 2025, and the subject area was restricted to Neuroscience, review articles were excluded from the results. Furthermore, eligibility was limited to articles published in open access format and written exclusively in English. Choosing these filters aimed to ensure the inclusion of recent, accessible, and relevant human studies pertinent to our research objectives.

The selection process started with title and abstract screening performed independently by two reviewers. After this first step, we retrieved the full texts of the potentially eligible studies and assessed them again, applying the predefined inclusion and exclusion criteria. Any disagreements were discussed and, if necessary, a third reviewer was consulted to reach consensus.

Data was extracted from each included study independently by two reviewers using a pre-defined data extraction table. The extracted variables included study characteristics, sample size, sex distribution (percentage of male participants), and the number of patients with hypertension, diabetes, obesity, dyslipidemia, smoking habits, chronic alcohol consumption, atrial fibrillation, and a history of transient ischemic attack (TIA). Additionally, we collected reported effect measures, such as odds ratios (ORs) with 95% confidence intervals (CI), for each risk factor when they were available. All extracted data were organized and stored using Microsoft Excel. ORs with 95% CI, as reported by the studies, were used as effect measure for each risk factor. When multiple effect estimates were reported, we prioritized fully adjusted models at the most recent point available. We did not use any automated tools or software to assist in the screening or selection process and did not contact study authors for additional information.

Studies that reported ORs for the association between risk factors and ischemic stroke were selected for synthesis, while those in which ORs were not clearly defined or lacked sufficient raw data to allow OR calculation were excluded. We did not perform any data conversions or transformations prior to synthesis. Data from the included studies were summarized in descriptive tables. Meta-analysis results were visually presented using forest plots showing individual and pooled effect estimates with their corresponding 95% CI. Meta-analyses were performed using a random-effects model, with restricted maximum-likelihood [REML] estimator, to account for between-study variability. Statistical heterogeneity was assessed using the I^2^ statistic, with values above 50% considered as substantial heterogeneity and with the Q-test assessing statistical significance (threshold of *p* < 0.05). All statistical analyses were performed using R software (version 4.4.3), with the ‘metafor’ package. We performed leave-one-out sensitivity analyses for each risk factor. For factors with substantial heterogeneity (I^2^ > 75%), we conducted exploratory subgroup analyses by primary outcome (incident vs. recurrent vs. post-stroke), study design (cohort vs. cross-sectional vs. retrospective/case–control), and region (Asia vs. Europe vs. North America) when ≥2 studies per stratum were available. Between-subgroup differences were tested using mixed-effects models (REML) with the subgroup as a moderator (Q_between). Publication bias was assessed by visual inspection of funnel plots and Egger’s regression test. For syntheses including ≤10 studies, these tests are known to be underpowered, and results should therefore be interpreted with caution.

In addition, the methodological quality of the included studies was assessed using the Newcastle–Ottawa Scale (NOS). Cohort and case–control studies were formally assessed using the appropriate versions of the Newcastle–Ottawa Scale (NOS). Purely cross-sectional prevalence studies were described narratively, as NOS criteria are not directly applicable to these designs. The detailed NOS scores are provided in Appendix A.

The certainty of evidence was assessed using the GRADE (Grading of Recommendations, Assessment, Development and Evaluation) framework. Because all included studies were observational in design, the certainty of evidence for each risk factor started at Low. Certainty was further downgraded when major limitations were identified across the following domains: risk of bias, inconsistency (heterogeneity), indirectness, imprecision, and publication bias. A Summary of Findings (SoF) table was constructed to present the pooled effect estimates, number of studies and participants, the final GRADE rating, and the reasons for downgrading.

## 3. Results

A total of 2910 records were identified through database searches. After removing duplicates, 2386 unique records were screened by title and abstract. Of these, 500 full-text articles were assessed for eligibility, resulting in 23 studies that met the inclusion criteria and were included in the review. Study identification, screening, eligibility assessment, and inclusion are summarized in the PRISMA 2020 flow diagram (Figure 1). Reasons for exclusion at the full-text stage included insufficient data for extraction, wrong study focus, and inappropriate outcome measurements. A detailed list of excluded studies with references is provided in Appendix A.

The list of included studies and their key characteristics is provided in Table 1. A full version with complete information on studies included is available as Appendix A.

We assessed the risk of small-study effects or publication bias for each meta-analysis using Egger’s regression test for funnel plot asymmetry. The results were not statistically significant for any of the analyzed risk factors: hypertension (*p* = 0.30), dyslipidemia (*p* = 0.49), smoking (*p* = 0.42), diabetes mellitus (*p* = 0.91), obesity (*p* = 0.49), atrial fibrillation (*p* = 0.96), and history of transient ischemic attack (*p* = 0.28). Funnel plots are available in the Appendix A.

The results of the included studies are summarized by risk factor. For each risk factor, pooled effect estimates (ORs with 95% CI) are reported. These are presented in Table 2 and Figure 2, Figure 3, Figure 4, Figure 5, Figure 6, Figure 7, Figure 8 and Figure 9 (forest plots).

The contributing studies varied in population characteristics, study settings, and sample sizes, which may contribute to heterogeneity.

We conducted random-effects meta-analyses for each investigated risk factor: hypertension, diabetes mellitus type 2, dyslipidemia, smoking, obesity, atrial fibrillation, and history of transient ischemic attack. Measures of statistical heterogeneity (I^2^, tau^2^, and *p*-value for Cochran’s Q test) were also presented. All forest plots included the direction of the effect, with OR values greater than 1 suggesting a higher likelihood of ischemic stroke associated with the corresponding risk factor.

Sensitivity analyses were performed by sequentially removing individual studies (leave-one-out analysis) to assess the stability of the results. These analyses showed that the overall findings remained consistent.

No subgroup stratification (e.g., by age or sex) was possible, as most studies reported only aggregated effect sizes.

Egger’s tests were non-significant across all risk factors. However, for syntheses including ≤10 studies (e.g., atrial fibrillation, TIA), the analyses were underpowered to reliably detect asymmetry; therefore, no clear evidence of small-study effects was observed.

Thirteen studies with statistical significant high heterogeneity—I^2^ = 91.72%, Q (12) = 50.86, *p* < 0.0001—were included in the meta-analysis to evaluate the association between hypertension and ischemic stroke. The corresponding forest plot is presented in Figure 2. Leave-one-out analyses did not materially change the estimate (e.g., excluding [51] → OR 1.47 [1.26–1.70]; excluding [54] → OR 1.51 [1.24–1.84]; excluding [38] → OR 1.64 [1.38–1.95]).

Exploratory subgroup analyses showed statistically significant between-subgroup heterogeneity. By primary outcome (Q_between = 9.24, *p* = 0.0099): incident stroke studies yielded OR 1.49 [1.20–1.86], I^2^ = 90.6%; post-stroke outcome studies OR 1.31 [1.13–1.52], I^2^ = 0%; recurrent stroke studies OR 3.21 [2.12–4.87], I^2^ = 0%. By design (Q_between = 8.21, *p* = 0.016): cohort studies OR 1.44 [1.24–1.69], I^2^ = 85.6%; cross-sectional studies OR 1.44 [0.94–2.22], I^2^ = 78.3%; retrospective studies OR 3.21 [2.12–4.87], I^2^ = 0%. By region (Q_between = 7.21, *p* = 0.027): Asian studies OR 1.31 [0.83–2.06], I^2^ = 81.9%; European studies OR 1.39 [1.22–1.57], I^2^ = 82.4%; North American studies OR 3.21 [2.12–4.87], I^2^ = 0%.

Notably, several strata included few studies (df = 1–4), so these subgroup findings should be interpreted with caution.

Eleven studies (Q (10) = 119.35, *p* < 0.0001, I^2^ = 96.03%) were included in the meta-analysis of dyslipidemia and ischemic stroke (Figure 3). Leave-one-out analyses did not materially change the overall estimate; the largest shifts occurred when excluding Chung JY et al. (OR 1.16 [0.95–1.42]), Poupore N et al. (OR 1.50 [0.98–2.31]), and Liu Z et al. (OR 1.45 [0.96–2.21]).

Subgroup analyses by primary outcome suggested borderline differences (Q_between = 3.50, *p* = 0.061). Incident stroke studies showed a consistent positive association (OR 1.30 [1.15–1.48]; I^2^ = 35.0%), whereas post-stroke outcome studies showed weaker and more heterogeneous estimates (OR 0.99 [0.69–1.40]; I^2^ = 70.7%).

By study design, no significant differences were detected (Q_between = 1.37, *p* = 0.504). Cohort studies indicated a modest but significant association (OR 1.25 [1.02–1.53]; I^2^ = 72.5%), cross-sectional studies showed a similar but non-significant estimate (OR 1.25 [0.94–1.66]; I^2^ = 33.3%), while retrospective studies yielded highly unstable results (OR 2.40 [0.22–26.18]; I^2^ = 99.0%), reflecting the influence of very few data points.

Regional subgroup analyses were not feasible due to insufficient studies per level.

We included a total of 9 studies in the meta-analysis to evaluate the association between smoking and ischemic stroke (Q (8) = 86.04, *p* < 0.0001, I^2^ = 94.07%). The results are shown in Figure 4. Leave-one-out checks did not materially change the result; the largest shifts occurred when excluding Chung JY et al. (OR 1.41 [1.19–1.66]), Samuthpongtorn C et al. (OR 1.73 [1.23–2.42]), and Tsai CF et al. (OR 1.71 [1.17–2.49]).

Subgroup by primary outcome showed no between-stratum differences (Q_between = 0.15, *p* = 0.697): incident stroke studies OR 1.45 [1.14–1.84]; I^2^ = 82.7% vs. post-stroke outcome studies OR 1.35 [1.11–1.65]; I^2^ = 0.0%.

Subgroup by design indicated significant heterogeneity across strata (Q_between = 6.81, *p* = 0.0091): cohort OR 1.18 [0.97–1.42]; I^2^ = 32.9% versus cross-sectional OR 1.56 [1.38–1.77]; I^2^ = 17.0%. Regional subgroup analyses were not feasible due to insufficient studies per level.

Thirteen studies (Q (12) = 326.65, I^2^ = 90.22%) were included in the meta-analysis to assess the association between diabetes mellitus type 2 and ischemic stroke (Figure 5). No genetic risk studies were included in the DM2 meta-analysis. Leave-one-out sensitivity checks showed that the association remained robust, with the largest shifts observed when excluding Liu C et al. (OR 1.64 [1.42–1.90]), Muhammad IF et al. (OR 1.47 [1.24–1.75]), and Johansson A et al. (OR 1.47 [1.24–1.75]).

Subgroup analyses by primary outcome suggested no statistically significant between-stratum heterogeneity (Q_between = 4.79, *p* = 0.091). Incident stroke studies showed a stronger association (OR 1.69 [1.38–2.07]; I^2^ = 81.5%), post-stroke outcome studies indicated a weaker and non-significant association (OR 1.19 [0.91–1.56]; I^2^ = 72.3%), while recurrent stroke studies demonstrated a robust effect (OR 1.69 [1.28–2.23]; I^2^ = 0.0%).

Subgroup analyses by study design also showed no significant differences between strata (Q_between = 3.44, *p* = 0.179). Cohort studies reported the strongest effect (OR 1.83 [1.51–2.21]; I^2^ = 78.0%), while cross-sectional (OR 1.37 [1.05–1.79]; I^2^ = 56.9%) and retrospective studies (OR 1.37 [0.88–2.13]; I^2^ = 86.3%) yielded weaker and less consistent estimates. Regional subgroup analyses were not feasible due to insufficient studies per level.

We included 7 studies (Q (6) = 13.04, I^2^ = 52.74%) in the meta-analysis that assessed the association between obesity and ischemic stroke. The results are shown in Figure 6. Nine studies (Q (8) = 46.34, I^2^ = 88.13%) were included in this meta-analysis to assess the association between alcohol consumption and ischemic stroke (Figure 7).

The meta-analysis of AF included 5 studies (Q (4) = 11.94, I^2^ = 66.75%). The results are shown in Figure 8.

We included three studies in the meta-analysis that evaluated the association between the history of TIA and ischemic stroke (Q (2) = 1.17, I^2^ = 0%). The results are shown in Figure 9.

The relative weight of individual risk factors is presented in Figure 10 and detailed in Table 2, providing a comparative overview of their contribution to ischemic stroke risk.

The certainty of evidence assessment is presented in Table 3 (Summary of Findings). All risk factors started at Low certainty, as all included studies were observational. Certainty was further downgraded for several factors due to inconsistency and/or imprecision.

These results provide a comprehensive overview of the main risk factors associated with ischemic stroke and form the basis for further interpretation in the context of existing literature. The following section discusses these findings in detail, highlighting their clinical relevance and potential implications.

## 4. Discussion

In this systematic review and meta-analysis, we found significant associations between ischemic stroke and hypertension, smoking, diabetes mellitus type 2, AF, and history of TIA. No statistically significant associations were found for dyslipidemia, obesity or chronic alcohol consumption, as their confidence intervals included the null effect.

An important consideration when interpreting these findings is the heterogeneity in operational definitions of exposures across the included studies. For example, smoking was variably classified as current versus ever-smoker, obesity was assessed using either body mass index (BMI) or waist-to-hip ratio, and dyslipidemia was defined as a composite clinical diagnosis or based on specific lipid fractions (LDL, HDL, triglycerides). Such variability likely contributed to the null associations observed for obesity and dyslipidemia (Table 2, Figure 10). In addition, the use of statin therapy in contemporary cohorts may have attenuated the apparent association with dyslipidemia, while the clustering of multiple risk factors within the metabolic syndrome may have mediated observed associations. These methodological differences complicate the interpretation of isolated effects and emphasize the need for standardized definitions in future research.

Among the investigated risk factors, hypertension, diabetes mellitus type 2, dyslipidemia, and smoking exhibited the highest heterogeneity (I^2^ > 90% for most analyses). Therefore, we performed additional sensitivity and exploratory subgroup analyses for these four factors to investigate potential sources of variability and to ensure the robustness of the pooled estimates. We excluded genetic risk studies from pooling; a single GRS study is described qualitatively.

Certainty of evidence was rated as low for most risk factors, including hypertension, dyslipidemia, smoking, diabetes mellitus type 2, AF, history of TIA and as very low for obesity and chronic alcohol consumption. This was primarily due to the lack of formal risk of bias assessment of studies included and the presence of substantial statistical heterogeneity for most syntheses, except for TIA, where no heterogeneity was detected. The effect estimates were generally precise, and no significant publication bias was identified using Egger’s test.

### 4.1. Hypertension, Ischemic Stroke and Neuroinflammation

Hypertension emerged as a robust and consistent risk factor for ischemic stroke, with sensitivity checks confirming the stability of the association. However, the high degree of heterogeneity (I^2^ > 90%) warrants careful interpretation. Subgroup analyses suggested that effect sizes varied by outcome and study design: recurrent stroke and retrospective studies yielded particularly large estimates, whereas post-stroke outcomes indicated more modest associations. Regional analyses also pointed to stronger effects in European and North American cohorts compared with Asian studies. These patterns likely reflect methodological and population differences, including variation in blood pressure definitions, treatment availability, and baseline cardiovascular risk profiles. While the overall evidence strongly supports hypertension as a major determinant of stroke, the heterogeneity observed highlights the need for standardized definitions and consistent adjustment for confounders in future research. While the strength of the association is moderate, the result supports the well-established role of hypertension as a major modifiable risk factor for stroke.

Our results are similar to those of previous studies that identified hypertension as a key determinant in the pathogenesis of ischemic stroke. For example, studies such as [57] agree that hypertension and its consequences are associated with more than 50% of ischemic strokes and 70% of hemorrhagic strokes. Despite good blood pressure control, there remains a 10% risk of recurrent cerebrovascular events, reinforcing the importance of blood pressure control in both primary and secondary stroke prevention. This hemodynamic strain is also closely linked to increased oxidative stress, which contributes to vascular injury and impaired autoregulation.

With age, arterial stiffness increases, sympathetic tone decreases, and autonomic dysfunction occurs, which directly leads to increased variations in blood pressure values. Therefore, it becomes increasingly difficult to maintain normal cerebral flow, thus increasing the risk of cerebral ischemia [57]. Hypertension was found to be associated with all subtypes of ischemic stroke, including cardioembolism related to AF [58], large artery stroke due to carotid stenosis, lacunar stroke, or less common etiologies of stroke, such as carotid dissection [59]. Mechanisms most frequently involved in the pathogenesis of ischemic stroke include lipohyalinosis in small, penetrating arteries and the decreased capacity for self-regulation of cerebral perfusion through the increasing variation in blood pressure values with age. Recent evidence has also highlighted the relevance of cerebral microbleeds as a novel finding associated with chronic hypertension [60]. These lesions may contribute to impaired brain homeostasis and increase vulnerability to ischemic injury, underscoring the multifaceted cerebrovascular impact of hypertension. Emerging research further implicates cerebral microbleeds as metabolically active lesions associated with hypertension-induced small vessel disease [61], reflective of chronic microvascular instability. Studies have suggested that intra-individual variability in blood pressure measurements, or differences in blood pressure measurements at different times within an individual, are associated with a lower stroke risk than elevated average blood pressure alone [62].

Hypertension is a major contributor to oxidative stress in the cerebral circulation. Activation of the renin–angiotensin–aldosterone system (RAAS), particularly through angiotensin II, stimulates the expression of NADPH oxidases (mainly NOX1, NOX2, and NOX5) in endothelial and vascular smooth muscle cells [63]. This leads to an overproduction of superoxide (O_2_^−^), which reacts with nitric oxide (NO) to form peroxynitrite (ONOO^−^), thereby reducing NO bioavailability and promoting endothelial dysfunction. Additionally, the disturbed shear stress associated with elevated blood pressure increases mitochondrial ROS production and uncouples endothelial nitric oxide synthase (eNOS), further amplifying vascular oxidative injury [64]. In the cerebral circulation, these processes contribute to arterial stiffening, small vessel disease, and an increased risk of lacunar infarctions and ischemic stroke [65].

The interplay between inflammation, oxidative stress, and hypertension can contribute to further endothelial damage, atherosclerosis acceleration, instability of the plaque, and thrombi embolic mechanisms [66].

### 4.2. Dyslipidemia, Ischemic Stroke and Neuroinflammation

Dyslipidemia was not significantly associated with ischemic stroke in our meta-analysis (Figure 3). For dyslipidemia, the overall association with ischemic stroke was inconsistent, with sensitivity checks confirming that no single study unduly influenced the results. Subgroup analyses suggested that the effect may depend on the type of outcome assessed: incident stroke studies indicated a modest but consistent positive association, whereas post-stroke outcome studies showed weaker and more heterogeneous effects. No significant differences emerged by study design, although cohort studies demonstrated a small but significant association, while retrospective analyses produced highly unstable estimates due to sparse data. Taken together, these findings highlight the uncertainty surrounding dyslipidemia as an independent risk factor for stroke, which may reflect variability in lipid measurement methods, treatment exposure (e.g., statins), and population characteristics across studies. Future research using harmonized definitions and adjustment for lipid-lowering therapy is needed to clarify this relationship. Definitions of dyslipidemia varied across the included studies, ranging from documented diagnosis or lipid-lowering therapy to explicit biochemical thresholds (see Appendix A). Although the pooled estimate did not reach statistical significance, incident stroke cohorts demonstrated consistent positive associations. The direction-of-effect therefore likely persists; however, widespread statin use and related treatment confounding in contemporary cohorts may have attenuated the observed association.

This differs from previous findings that reported a positive association, particularly in relation to elevated LDL cholesterol or low HDL cholesterol levels [8]. The discrepancy with previous studies may reflect differences in lipid profile characterization (e.g., LDL vs. HDL cholesterol), population-specific genetic backgrounds, dietary patterns, or study design heterogeneity. Additionally, the inconsistent definitions of dyslipidemia across studies may have diluted the observed effect.

An altered lipid profile is considered to accelerate the process of atherosclerosis, thus increasing the risk of developing an ischemic stroke by atherothrombotic or lacunar mechanisms through endothelial dysfunction, initially, and then through the development of atherosclerotic plaques, stenosis, or vessel occlusion [67]. The pro-atherogenic environment is reinforced by oxidative stress, further promoting endothelial dysfunction.

Small dense LDL (smLDL) particles are particularly prone to oxidative modification, generating oxidized LDL (ox-LDL) that promotes endothelial dysfunction by activating NADPH oxidase and uncoupling eNOS, thus increasing superoxide production and reducing NO bioavailability [68]. Moreover, ox-LDL encourages foam cell formation via macrophage uptake and stimulates pro-inflammatory signaling pathways (e.g., NF-κB, MAPK), fostering atherosclerotic plaque development. Remnant lipoproteins (from VLDL/TG-rich lipoproteins) contribute further by promoting endothelial ROS production and directly depositing cholesterol within vessel walls [68]. Additionally, oxysterols such as 7-ketocholesterol—oxidative derivatives of cholesterol—induce mitochondrial dysfunction, endothelial apoptosis, and inflammasome activation, thereby destabilizing plaques [69]. Together, these oxidative mechanisms accelerate atherogenesis, impair cerebral microvasculature, and heighten the risk of cerebral infarctions and ischemic stroke. Considering the established association between dyslipidemia and systemic inflammation [70], it is plausible that this pro-inflammatory state extends to the central nervous system, thereby contributing to neuroinflammation and potentially exacerbating neurological disorders.

### 4.3. Diabetes Mellitus Type 2, Ischemic Stroke and Neuroinflammation

Type 2 diabetes mellitus was consistently associated with an increased risk of ischemic stroke, and sensitivity analyses confirmed the robustness of this relationship. Although heterogeneity was high overall (I^2^ > 90%), subgroup analyses did not identify statistically significant between-stratum differences. Still, some patterns emerged: incident and recurrent stroke studies indicated stronger associations, whereas post-stroke outcome studies suggested weaker and less consistent effects. Similarly, cohort studies yielded the largest pooled effect, while cross-sectional and retrospective designs showed attenuated estimates with greater variability. Regional analyses were not feasible, limiting broader interpretation. Taken together, these findings underscore the role of diabetes as an established risk factor for stroke, while also reflecting methodological and clinical diversity across available studies. The results indicate that individuals with diabetes have 53% higher odds of experiencing ischemic stroke compared to non-diabetic individuals, highlighting the substantial cerebrovascular risk posed by this metabolic disorder. Our results were in accordance with several studies suggesting that patients with diabetes have a two times higher risk of developing ischemic stroke [71].

Diabetes mellitus, especially type 2, alters the proper functioning of energetic metabolism, resulting not only in higher glycemic values, but also in a modified lipid profile, enhancing the atherosclerosis process. The increasing prevalence of diabetes mellitus may partially explain the risk of stroke in younger populations [72]. Glycemic control alone in diabetics does not confer a lower risk than intensive behavior modification in association with medical intervention [73]. The observed association may be explained by several pathophysiological mechanisms, including chronic hyperglycemia-induced endothelial dysfunction, accelerated atherosclerosis, increased platelet aggregation, and a pro-inflammatory state. These metabolic alterations amplify oxidative stress, which accelerates vascular damage in diabetic patients [74].

Diabetes mellitus promotes oxidative stress through multiple interlinked mechanisms, particularly involving pancreatic β-cell dysfunction, insulin resistance, and mitochondrial overactivation [75]. Hyperglycemia and hyperlipidemia increase glycolytic flux and tricarboxylic acid (TCA) cycle activity in β-cells, leading to elevated ATP production and mitochondrial ROS generation [76]. Moreover, intracellular calcium influx activates protein kinase C (PKC), which stimulates superoxide production via NADPH oxidase pathways [76]. The resulting oxidative stress impairs insulin secretion, contributing to chronic hyperglycemia and further ROS accumulation—establishing a vicious cycle. In parallel, peripheral insulin resistance worsens glucose metabolism and enhances oxidative injury to endothelial cells. These changes accelerate vascular inflammation, endothelial dysfunction, and prothrombotic states, all of which significantly increase the risk of ischemic stroke in diabetic patients [75].

### 4.4. Chronic Alcohol Consumption, Ischemic Stroke and Neuroinflammation

Chronic alcohol consumption was not statistically significant associated with ischemic stroke in our meta-analysis. The wide confidence interval crossing the null value suggests that the evidence is insufficient to support a definitive link between chronic alcohol use and ischemic stroke risk.

There has been evidence concerning the relationship between alcohol consumption and the risk of ischemic stroke, as follows: light to moderate alcohol consumption (up to two drinks per day in men and up to one drink per day in women) is protective against stroke, with excessive alcohol consumption being associated with an increased risk of stroke [77]. Excessive alcohol consumption is a risk factor for hypertension and is responsible for poor blood pressure control in patients with hypertension [78].

Chronic alcohol consumption significantly contributes to oxidative stress through both hepatic and extrahepatic mechanisms. In the liver, ethanol is oxidized to acetaldehyde by alcohol dehydrogenase (ADH), and subsequently to acetate by mitochondrial aldehyde dehydrogenase 2 (ALDH2), generating NADH and altering the cellular redox balance. At higher concentrations, chronic ethanol intake induces cytochrome P450 2E1 (CYP2E1), which metabolizes ethanol using NADPH and produces superoxide anions, leading to elevated levels of ROS and RNS [79]. In the brain, classical ADH-mediated pathways are virtually absent, but chronic alcohol exposure induces mitochondrial dysfunction, including impaired oxidative phosphorylation and disrupted dynamics of mitochondrial fission and fusion [80]. These changes result in excessive electron leakage from the electron transport chain and enhanced ROS generation. Accumulated oxidative damage in both hepatic and cerebral tissues promotes systemic inflammation, endothelial dysfunction, and neurovascular injury, which collectively increase the risk of ischemic stroke [81].

As a promoter of neuroinflammation, chronic alcohol consumption induces microglial activation and peripheral macrophage infiltration in the central nervous system, enhancing its contributing role in ischemic injuries [82].

### 4.5. Smoking, Ischemic Stroke and Neuroinflammation

Smoking was consistently associated with an increased risk of ischemic stroke, and the overall effect remained stable in sensitivity analyses. Despite the robust pooled association, heterogeneity was substantial (I^2^ > 90%). Subgroup analyses did not reveal differences between incident and post-stroke outcomes, suggesting that the detrimental effect of smoking applies across clinical contexts. In contrast, significant heterogeneity was observed by study design, with cross-sectional studies showing stronger associations than cohort studies, which may reflect differences in case ascertainment and the potential for recall or selection bias. The inability to perform regional subgroup analyses limits broader generalizability. Overall, these findings reinforce smoking as a major modifiable risk factor for stroke, while also highlighting methodological influences that can shape effect estimates. Studies suggest that smoking remains a major stroke risk factor, almost doubling the risk with a dose-dependent relationship with the pack-year index [83]. Smoking, through absorbed nicotine, produces inflammation of the vascular endothelium, subsequently leading to endothelial dysfunction, thus contributing to the acceleration of the atherosclerosis process. At the same time, it increases the degree of LDL cholesterol reactivity, contributing to the instability of atherosclerotic plaques with a higher risk of thrombus formation. Smoking is estimated to contribute to almost 15% of all stroke deaths annually [84]. The oxidative burden imposed by smoking overwhelms endogenous antioxidant defenses, leading to sustained vascular inflammation.

Cigarette smoking is a well-established trigger of oxidative stress, contributing significantly to vascular injury and ischemic stroke risk. Tobacco smoke contains over 4000 chemical compounds, including reactive oxidants and free radicals capable of entering the bloodstream and inducing macromolecular damage within endothelial cells. These reactive species not only exert direct oxidative effects but also activate NADPH oxidase and other ROS-generating enzymatic systems in the vascular wall, despite the exact components responsible remaining unclear [85]. Additionally, smoking elicits a systemic inflammatory response and promotes platelet activation, both of which further contribute to oxidative vascular stress and endothelial dysfunction. The resulting damage enhances atherogenesis, disrupts vascular homeostasis, and promotes thrombotic events—creating a pro-ischemic state particularly detrimental to cerebral circulation [86].

Several biopsies of the brain were obtained and analyzed from animal models exposed to nicotine. It was confirmed that pro-oxidant markers had increased expression in rats exposed to nicotine [87], thus reinforcing the hypothesis that nicotine-induced oxidative stress contributes to neuroinflammation and further to the development and progression of ischemic stroke.

### 4.6. Obesity, Ischemic Stroke and Neuroinflammation

Obesity was not significantly associated with ischemic stroke in our meta-analysis. Although the point estimate is close to the null value, the wide confidence interval suggests uncertainty regarding the true direction and magnitude of the association. One study showed that a sedentary lifestyle is closely related to an improper diet rich in carbohydrates and salt [88]. In subjects living a sedentary lifestyle, calories obtained through food are stored as lipid deposits, contributing to the process of atherosclerosis. At the same time, excess adipose tissue (overweight/obesity), especially in the abdomen, confers a pro-coagulant state to the patient, with an increased risk of stroke [89,90]. In the 2010 INTERSTROKE study, waist-to-hip ratio size was shown to be associated with a higher risk of stroke, unlike body mass index (BMI), which is commonly used in clinical practice [91].

The lack of significant findings may be partly due to the complex interplay between obesity and other metabolic risk factors such as hypertension, diabetes, and dyslipidemia, which could act as mediators or confounders [92,93]. Additionally, variability in the criteria used to define obesity (e.g., BMI thresholds) across studies may have contributed to inconsistent results.

Obesity induces a chronic inflammatory state, which contributes to neurodegeneration and cognitive decline. Together with its metabolic effects, obesity plays a major role in neuroinflammation as a risk factor for ischemic stroke and worsens the prognosis of post-stroke patients [94]. Excessive nutrient availability leads to mitochondrial overload, electron leakage, and overproduction of ROS, primarily via complexes I–III of the electron transport chain. Adipocyte hypertrophy increases the expression of NADPH oxidase isoforms (NOX2 in macrophages and NOX4 in adipocytes), which catalyze the formation of superoxide and hydrogen peroxide [95]. Free fatty acids also stimulate Toll-like receptor (TLR) signaling, activating NF-κB and further enhancing ROS production and cytokine release (e.g., TNF-α, IL-6). Additionally, endoplasmic reticulum (ER) stress and disrupted calcium homeostasis contribute to mitochondrial dysfunction and exacerbate ROS accumulation. These processes impair adiponectin secretion, promote leptin resistance, and fuel a systemic pro-inflammatory and pro-thrombotic state [95]. The resulting endothelial dysfunction, vascular remodeling, and altered energy homeostasis in the brain create a high-risk environment for ischemic stroke development.

### 4.7. Atrial Fibrillation, Ischemic Stroke and Neuroinflammation

AF was significantly associated with an increased risk of ischemic stroke, indicating that individuals with AF have 87% higher odds of experiencing ischemic stroke compared to those without AF, supporting the well-established role of cardiac arrhythmias in embolic stroke pathogenesis.

Oxidative stress has been implicated in atrial remodeling and endothelial dysfunction, providing a mechanistic link to stroke risk. The underlying mechanism is primarily related to blood stasis in the atria due to ineffective atrial contraction, which promotes thrombus formation and subsequent embolization of the cerebral arteries, leading to occlusion [96].

Atrial fibrillation is not directly related to a neuroinflammatory response, but the long-term neurodegeneration process [97], caused by microthrombi migration, promotes lower levels of brain perfusion in targeted areas, where oxidative stress molecules accumulate, therefore maintaining chronic inflammation.

### 4.8. History of TIA and Ischemic Stroke

A history of TIA was significantly associated with an increased risk of ischemic stroke in our meta-analysis, indicating that individuals with a prior TIA have 62% higher odds of developing ischemic stroke. This result highlights the prognostic importance of TIA as a warning event for future cerebrovascular accidents. The transient ischemia itself triggers bursts of oxidative stress, which may contribute to subsequent cerebrovascular vulnerability.

Overall, we identified several modifiable risk factors that were significantly associated with ischemic stroke, all of which converge mechanistically on oxidative stress and neuroinflammatory pathways.

Taken together, our findings contribute three main elements to the field. First, by restricting our analysis to studies published within the last five years, we demonstrate that the dominant modifiable risk factors for ischemic stroke have remained largely unchanged, underlining their persistent relevance for public health. Second, we integrate these epidemiological observations with mechanistic evidence on oxidative stress and neuroinflammation, offering a unified explanatory framework for how vascular risk factors promote cerebrovascular injury. Third, we highlight the implications for practice, including the need for personalized prevention and rehabilitation programs, and we suggest that antioxidant-based interventions may represent a promising complementary approach. These contributions strengthen the novelty of our work and provide directions for future research.

#### 4.8.1. Completeness and Applicability of Evidence

While our meta-analysis represents key factors commonly reported in the literature (e.g., hypertension, dyslipidemia, diabetes mellitus type 2, etc.), other potentially relevant risk factors, such as physical inactivity or diet, were not included due to lack of eligible studies meeting the inclusion criteria. The findings are applicable to adult populations at risk of ischemic stroke but may not fully represent all demographic or geographic contexts.

#### 4.8.2. Dominance of Research Locations

Most of the included studies were conducted in Asian and European populations, with fewer studies from America, or Africa. This geographical concentration may limit the generalizability of the findings to populations outside these regions, where genetic, environmental, and healthcare system differences might influence the observed associations.

#### 4.8.3. Participant Representativeness

The included studies involved adult patients admitted in hospitals, which may not fully represent the general population. As a result, the findings may be more applicable to individuals already receiving medical care or diagnosed with risk factors, rather than to community-based or asymptomatic populations.

#### 4.8.4. Participant Variation

Participant characteristics varied across studies in terms of age, sex distribution, and clinical profiles.

#### 4.8.5. Quality of the Evidence

When applying the GRADE framework, most associations between traditional vascular risk factors and ischemic stroke were rated as having Low to Very Low certainty of evidence. This reflects the observational nature of the available studies, the substantial heterogeneity across populations, and the imprecision of effect estimates for certain risk factors. Consequently, while our findings support the role of hypertension, diabetes, smoking, atrial fibrillation, and prior TIA as important contributors to ischemic stroke risk, the overall strength of evidence remains limited. These results highlight the need for more standardized, large-scale prospective studies to strengthen causal inference and improve the certainty of evidence regarding modifiable risk factors.

#### 4.8.6. Potential Biases in the Review Process

The search was limited to four major databases (PubMed, PubMed Central, Web of Science, and Scopus), and no additional searches were conducted in gray literature sources, which may have led to the omission of relevant studies. In addition, we restricted inclusion to studies published in the last 5 years, written in English, and available in open-access format. These criteria may have introduced selection bias by omitting relevant evidence outside this time frame, language, or accessibility filter. However, to reduce selection bias, study selection was conducted independently by two reviewers.

#### 4.8.7. Study Limitations

There are several limitations to our study. We acknowledge that non-modifiable contributors such as aging or genetic predispositions, as well as underexplored factors such as viral or bacterial infections, were not included in our meta-analysis. This deliberate restriction was driven by our aim to focus on modifiable and clinically actionable risk factors, supported by sufficient and comparable evidence across recent studies. The analyses were based on observational data, which limited the ability to establish causal relationships between the investigated risk factors and ischemic stroke. Another limitation is that included studies were heterogeneous in terms of their primary outcome, with some addressing incident ischemic stroke risk in the general population, others focusing on recurrent stroke, and a few reporting post-stroke outcomes. While we pooled results under the rationale of shared vascular risk mechanisms, this may have introduced additional heterogeneity and reduced the specificity of our estimates. Second, substantial heterogeneity was observed in several meta-analyses (e.g., for diabetes, dyslipidemia, and alcoholism), which may reduce the reliability of the pooled estimates. This heterogeneity likely reflects differences in the study design, sample characteristics, definitions of risk factors, and outcome ascertainment. Although leave-one-out and exploratory subgroup checks were performed, several strata contained few studies; therefore, residual heterogeneity remained substantial and subgroup signals should be interpreted with caution.

The number of studies included for some risk factors (e.g., atrial fibrillation, smoking, obesity, and history of TIA) was relatively small, which may have limited the statistical power and generalizability of these results. Additionally, we were unable to account for potential confounding factors such as medication use, socioeconomic status, or genetic predispositions due to the lack of access to individual-level data.

Another limitation is that cross-sectional studies could not be formally assessed with NOS and were therefore only narratively appraised.

Finally, publication bias cannot be ruled out, as studies with null or negative findings are less likely to be published. Although efforts were made to conduct a comprehensive literature search, it is possible that some relevant studies were missing.

However, the high levels of heterogeneity observed in studies assessing these factors suggest that the relationship may be context-dependent, possibly influenced by variations in study design, population characteristics, and risk factor definitions. These findings call for careful interpretation and emphasize the need for more standardized research in these areas. Our results support a multifaceted approach to stroke prevention, emphasizing the importance of integrated risk factor management. Future research should aim to further elucidate the interplay between individual and combined risk factors, ideally through high-quality prospective studies with access to patient-level data.

## 5. Conclusions

Our systematic review and meta-analysis confirm that the major modifiable risk factors for ischemic stroke—hypertension, type 2 diabetes mellitus, atrial fibrillation, and smoking—remain consistent determinants of risk, even in studies published within the last five years. This persistence underscores the continued need to address these factors as cornerstones of stroke prevention. Although obesity and chronic alcohol consumption did not reach statistical significance, their potential contribution cannot be excluded, given the heterogeneity of study populations and methodologies.

Importantly, converging evidence indicates that oxidative stress and neuroinflammation represent shared biological pathways through which these risk factors exert their deleterious effects on cerebrovascular health. Recognizing this unifying mechanism provides not only an explanatory framework but also practical implications for clinical care. Personalized rehabilitation programs integrating management of overlapping risk factors—such as combining cardiovascular monitoring with glycemic control, anticoagulation, or smoking cessation support—may improve long-term outcomes and reduce recurrence. Furthermore, as pro-oxidant processes mediate much of the vascular injury, antioxidant-based interventions may be biologically plausible and warrant exploration as complementary strategies to established prevention and rehabilitation approaches.

Future studies should clarify the contribution of less consistently reported risk factors and further explore how modulation of oxidative and inflammatory pathways may enhance both prevention and recovery in ischemic stroke.

### Clinical Translation Box

The findings of this review support integrated risk-factor management as the cornerstone of stroke prevention. In practice, this entails combining evidence-based interventions such as anticoagulation for atrial fibrillation, strict blood pressure and glycemic control, lipid-lowering therapy, and smoking cessation. Such multimodal strategies not only reduce recurrent events but also target the shared redox-inflammatory mechanisms that underpin cerebrovascular injury.

## Figures and Tables

**Figure 1 antioxidants-14-01229-f001:**
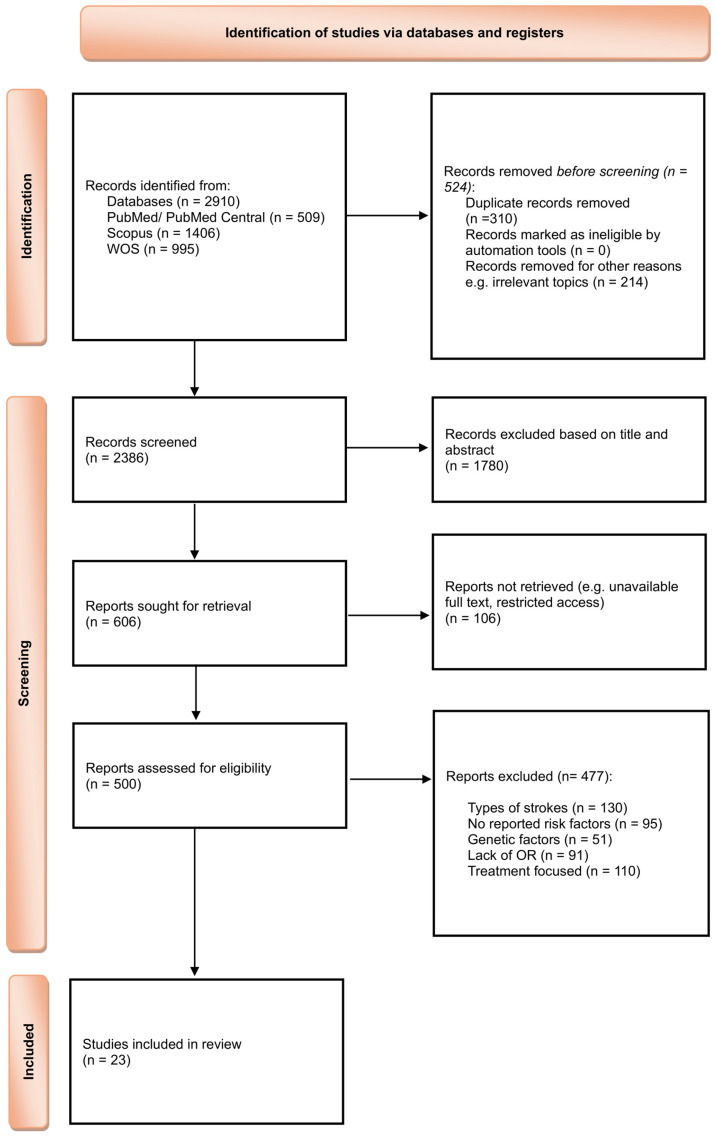
PRISMA 2020 flow diagram illustrating study selection process adapted from the official template (CC BY 4.0 license) [33].

**Figure 2 antioxidants-14-01229-f002:**
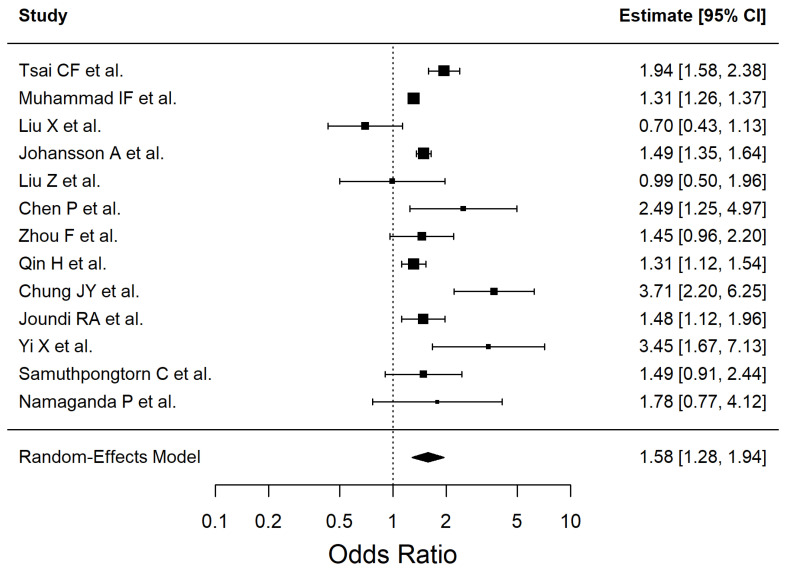
Forest plot showing the association between hypertension and ischemic stroke risk. Each square represents the study-specific odds ratio, with its size proportional to the study weight. Horizontal lines indicate the 95% confidence intervals. The vertical dashed line represents the null effect (OR = 1). The diamond indicates the pooled odds ratio and its 95% CI, estimated using a random-effects model. Studies included in this analysis are listed in the figure and correspond to references [34,35,38,40,41,43,44,50,51,52,54,55,56].

**Figure 3 antioxidants-14-01229-f003:**
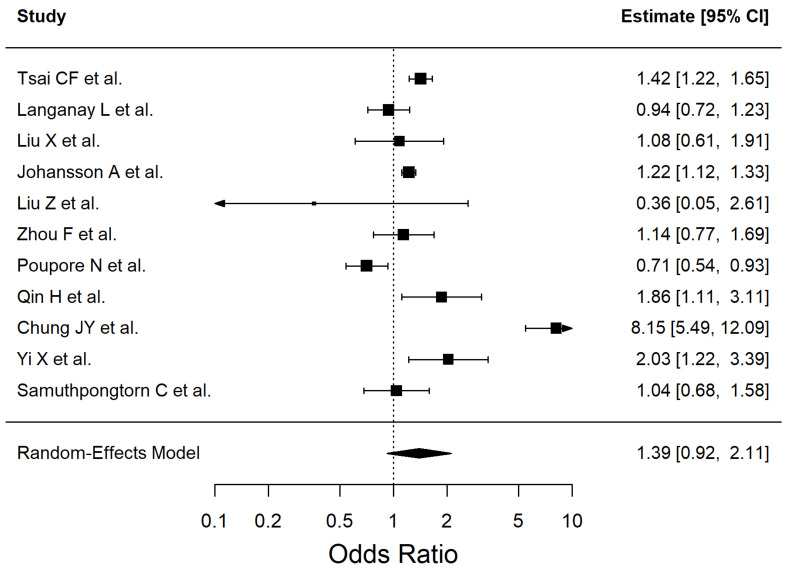
Forest plot showing the association between dyslipidemia and ischemic stroke risk. Each square represents the study-specific odds ratio, with its size proportional to the study weight. Horizontal lines indicate the 95% confidence intervals. The vertical dashed line represents the null effect (OR = 1). The diamond indicates the pooled odds ratio and its 95% CI, estimated using a random-effects model. Studies included in this analysis correspond to references [34,37,38,40,41,44,47,50,51,54,55].

**Figure 4 antioxidants-14-01229-f004:**
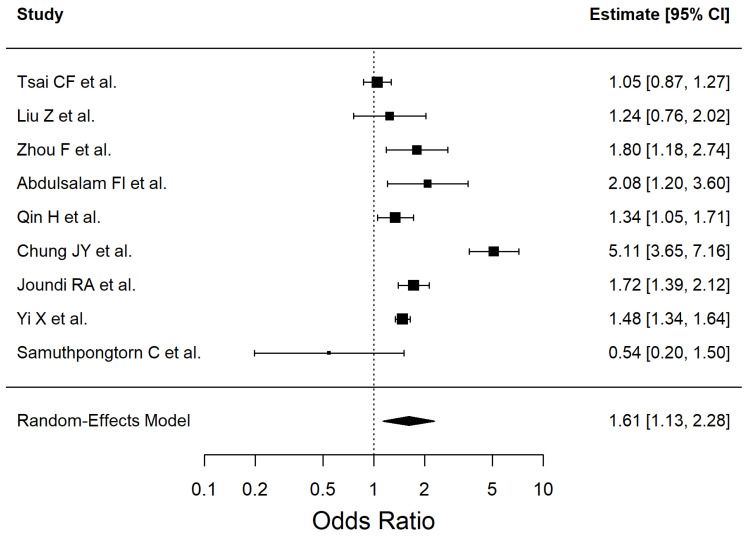
Forest plot showing the association between smoking and ischemic stroke risk. Each square represents the study-specific odds ratio, with its size proportional to the study weight. Horizontal lines indicate the 95% confidence intervals. The vertical dashed line represents the null effect (OR = 1). The diamond indicates the pooled odds ratio and its 95% CI, estimated using a random-effects model. Studies included in this analysis correspond to references [34,41,44,49,50,51,52,54,55].

**Figure 5 antioxidants-14-01229-f005:**
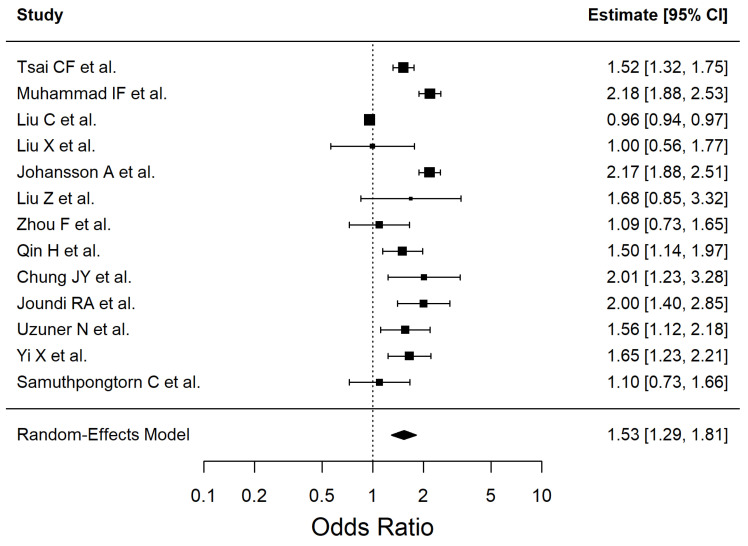
Forest plot showing the association between type 2 diabetes mellitus and ischemic stroke risk. Each square represents the study-specific odds ratio, with its size proportional to the study weight. Horizontal lines indicate the 95% confidence intervals. The vertical dashed line represents the null effect (OR = 1). The diamond indicates the pooled odds ratio and its 95% CI, estimated using a random-effects model. Studies included in this analysis correspond to references [34,35,36,38,40,41,44,50,51,52,53,54,55].

**Figure 6 antioxidants-14-01229-f006:**
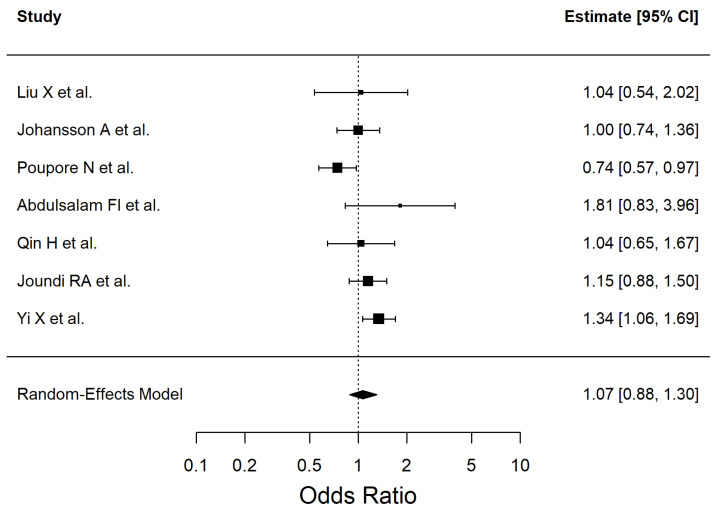
Forest plot showing the association between obesity and ischemic stroke risk. Each square represents the study-specific odds ratio, with its size proportional to the study weight. Horizontal lines indicate the 95% confidence intervals. The vertical dashed line represents the null effect (OR = 1). The diamond indicates the pooled odds ratio and its 95% CI, estimated using a random-effects model. Studies included in this analysis correspond to references [38,40,47,49,50,52,54].

**Figure 7 antioxidants-14-01229-f007:**
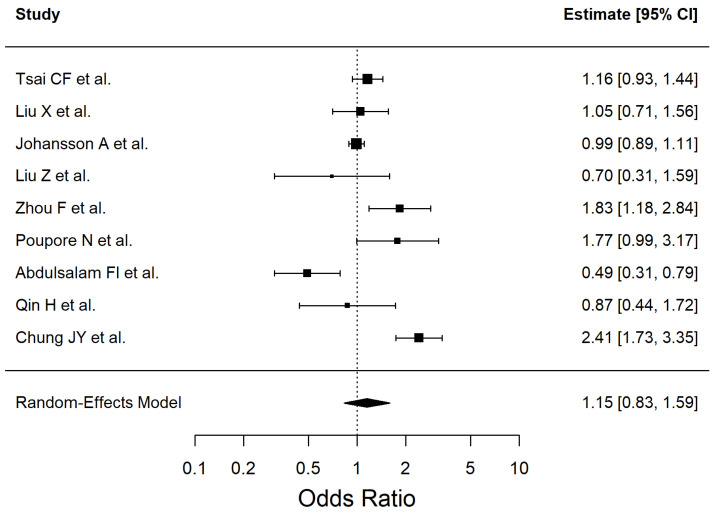
Forest plot showing the association between chronic alcohol consumption and ischemic stroke risk. Each square represents the study-specific odds ratio, with its size proportional to the study weight. Horizontal lines indicate the 95% confidence intervals. The vertical dashed line represents the null effect (OR = 1). The diamond indicates the pooled odds ratio and its 95% CI, estimated using a random-effects model. Studies included in this analysis correspond to references [34,38,40,41,44,47,49,50,51].

**Figure 8 antioxidants-14-01229-f008:**
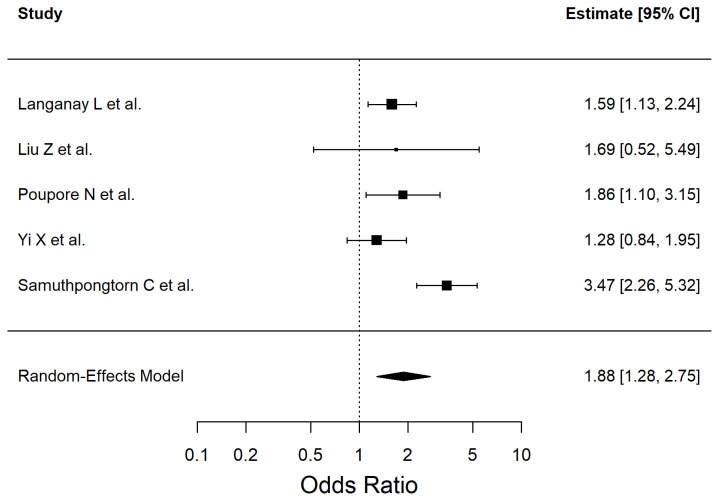
Forest plot showing the association between atrial fibrillation and ischemic stroke risk. Each square represents the study-specific odds ratio, with its size proportional to the study weight. Horizontal lines indicate the 95% confidence intervals. The vertical dashed line represents the null effect (OR = 1). The diamond indicates the pooled odds ratio and its 95% CI, estimated using a random-effects model. Studies included in this analysis correspond to references [37,41,47,54,55].

**Figure 9 antioxidants-14-01229-f009:**
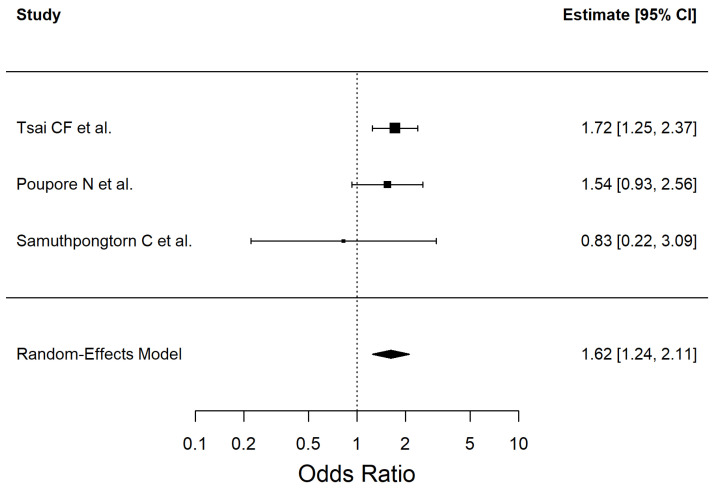
Forest plot showing the association between transient ischemic attack and ischemic stroke risk. Each square represents the study-specific odds ratio, with its size proportional to the study weight. Horizontal lines indicate the 95% confidence intervals. The vertical dashed line represents the null effect (OR = 1). The diamond indicates the pooled odds ratio and its 95% CI, estimated using a random-effects model. Studies included in this analysis correspond to references [34,47,55].

**Figure 10 antioxidants-14-01229-f010:**
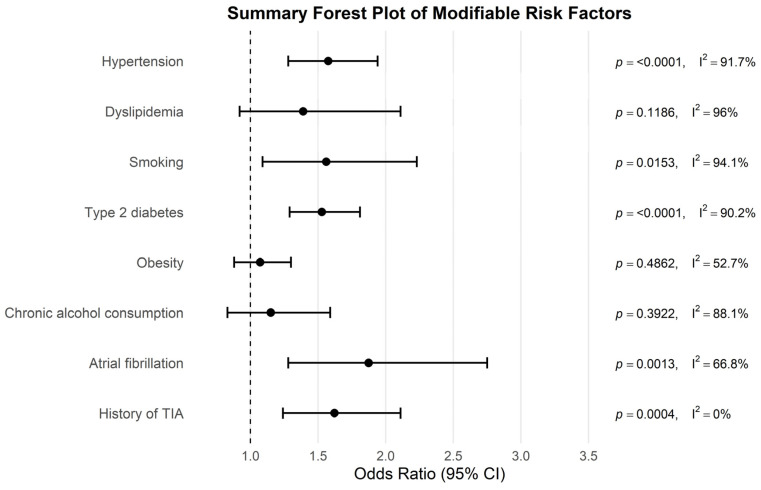
Summary forest plot showing the pooled odds ratios (ORs) and 95% confidence intervals for the risk factors associated with ischemic stroke.

**Table 1 antioxidants-14-01229-t001:** Summary of the studies included in the analysis.

Nr. Crt.	Study	Country	Study Design	Sample Size	Primary Outcome	Key Risk Factors *
1.	Tsai CF et al., 2021 [34]	Taiwan	Cohort	4953	Incident stroke	HT, DL, SM, DM2, ALC, TIA
2.	Muhammad IF et al., 2021 [35]	Sweden	Cohort	1838	Incident stroke	HT, SM, DM 2
3.	Liu C et al., 2024 [36]	China	Retrospective	24,355	Post-stroke outcome	DM 2
4.	Langanay L et al., 2024 [37]	France	Cohort	1912	Post-stroke outcome	DL, AF
5.	Liu X et al., 2024 [38]	China	Cross-sectional	504	Incident stroke	HT, DL, DM2, OB, ALC
6.	Zheng H et al., 2023 [39]	China	Cohort	54,123	Incident stroke	HT, SM, DM2, ALC
7.	Johansson A et al., 2021 [40]	Sweden	Cohort	26,549	Incident stroke	HT, DL, DM2, OB, ALC,
8.	Liu Z et al., 2020 [41]	China	Cohort	1121	Post-stroke outcome	HT, DL, SM, DM2, ALC, AF
9.	Ma F et al., 2023 [42]	China	Retrospective	190	Post-stroke outcome	HT, DM2
10.	Chen P et al., 2024 [43]	USA	Retrospective	51	Recurrent stroke	HT
11.	Zhou F et al., 2020 [44]	China	Cross-sectional	1101	Incident stroke	HT, DL, SM, DM2, ALC
12.	Dittrich TD et al., 2024 [45]	Switzerland	Cohort	3995	Post-stroke outcome	HT, DL, DM2
13.	Marston NA et al., 2022 [46]	International	Cohort	51,288	Incident stroke	Polygenic risk score (GRS); descriptive only, not pooled.
14.	Poupore N et al., 2020 [47]	USA	Retrospective	5469	Post-stroke outcome	DL, OB, ALC, AF, TIA
15.	Zhang G et al., 2025 [48]	China	Cross-sectional	5816	Other (stenosis phenotype)	HT, DL, DM2
16.	Abdulsalam FI et al., 2024 [49]	Thailand	Cross-sectional	708	Post-stroke outcome	SM, OB, ALC
17.	Qin H et al., 2020 [50]	China	Cohort	12,415	Post-stroke outcome	HT, DL, SM, DM 2, OB, ALC
18.	Chung JY et al., 2023 [51]	USA	Retrospective	787	Recurrent stroke	HT, DL, SM, DM2, ALC
19.	Joundi RA et al., 2022 [52]	Canada	Cross-sectional	492,400	Incident stroke	HT, SM, DM2, OB, ALC
20.	Uzuner N et al., 2023 [53]	Turkey	Retrospective	927	Recurrent stroke	DM 2
21.	Yi X et al., 2020 [54]	China	Cross-sectional	429	Incident stroke	HT, SM, DM2, OB, AF
22.	Samuthpongtorn C et al., 2021 [55]	Thailand	Cross-sectional	542	Post-stroke outcome	HT, DL, SM, DM2, AF, TIA
23.	Namaganda P et al., 2022 [56]	Uganda	Case–control	51	Incident stroke	HT

* HT—Hypertension, DL—Dyslipidemia, SM—Smoking, ALC—Chronic alcohol consumption, DM2—Diabetes mellitus type 2, OB—Obesity, AF—Atrial fibrillation, TIA—History of TIA.

**Table 2 antioxidants-14-01229-t002:** Pooled effect estimates for each risk factor identified.

Risk Factor	Pooled OR	95% CI	*p*-Value	I^2^%
Hypertension	1.575	1.28–1.94	<0.0001	91.72
Dyslipidemia	1.39	0.92–2.11	0.1186	96.03
Smoking	1.56	1.09–2.23	0.0153	94.07
Diabetes mellitus type 2	1.528	1.29–1.81	<0.0001	90.22
Obesity	1.072	0.88–1.30	0.4862	52.74
Chronic alcohol consumption	1.15	0.83–1.59	0.3922	88.13
Atrial fibrillation	1.874	1.28–2.75	0.0013	66.75
History of TIA	1.621	1.24–2.11	0.0004	0

**Table 3 antioxidants-14-01229-t003:** Summary of Findings on all risk factors.

Risk Factor	Effect Estimate (OR, 95% CI)	No. of Studies	No. of Participants	Certainty of Evidence (GRADE)	Reasons for Downgrade (If Any)
Hypertension	1.58 (1.28–1.94)	13	542,741	Very low	Serious inconsistency (I^2^ = 91.7%), Study design (observational evidence)
Diabetes mellitus type 2	1.53 (1.29–1.81)	13	567,921	Low	Inconsistency (I^2^ = 90%), Study design (observational evidence)
Dyslipidemia	1.39 (0.92–2.11)	11	55,782	Very low	Inconsistency (I^2^ = 96%), Imprecision (95% CI crosses null effect), Study design (observational evidence)
Smoking	1.56 (1.09–2.23)	9	514,456	Low	Inconsistency (I^2^ = 94%), Study design (observational evidence)
Obesity	1.07 (0.88–1.30)	6	526,059	Very low	Imprecision (95% CI crosses null effect), Study design (observational evidence)
Chronic alcohol consumption	1.15 (0.83–1.59)	9	53,607	Very low	Inconsistency (I^2^ = 88.13%), Imprecision (CI include 1), Study design (observational evidence)
Atrial Fibrillation	1.87 (1.28–2.75)	5	9473	Low	Inconsistency (I^2^ = 66.75%), Study design (observational evidence)
History of TIA	1.62 (1.24–2.11)	3	10,964	Low	Study design (observational evidence)

## Data Availability

All data used in this study are from previously published sources, as cited in the reference list. No new data was generated.

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
