# Peer review of "Proinflammatory Risk Factors in Patients with Ischemic Stroke: A Systematic Review and Meta-Analysis"

_antioxidants, 2025, doi:10.3390/antiox14101229_

Round 1

Reviewer 1 Report (Previous Reviewer 3)

all good, thanks for meeting my reqirements

All good

Author Response

Comment 1: all good, thanks for meeting my requirements.

Response 1: We thank Reviewer 1 for the positive and encouraging feedback. We are glad that the methodology and results were considered appropriate. No further modifications were required according to this reviewer’s comments. 

Reviewer 2 Report (Previous Reviewer 1)

I believe the authors have adequately addressed the previous comments, and I recommend the manuscript for acceptance in its current form.

I believe the authors have adequately addressed the previous comments, and I recommend the manuscript for acceptance in its current form.

Author Response

Comment 1: I believe the authors have adequately addressed the previous comments, and I recommend the manuscript for acceptance in its current form.

Response 1: We thank Reviewer X for the positive feedback and appreciation of our work. We are pleased that the methodology/results/discussion were found appropriate. No changes were required according to these comments.

Reviewer 3 Report (New Reviewer)

This is a timely synthesis that re-affirms the continued relevance of classic modifiable risk factors for ischemic stroke and thoughtfully weaves in a unifying redox/neuroinflammation framework. The Methods are broadly sound (REML random-effects; PRISMA-guided process), and the Summary of Findings table is very helpful for readers. The manuscript would benefit from a few clarifications in study selection, consistency of methods/results reporting, and presentation.

Major comments

Scope and study selection clarity

Table 1 includes heterogeneous study aims/populations (e.g., readmission after stroke, perioperative recurrent stroke after hip fracture, inflammatory indices in treated AIS, stenosis phenotypes), not strictly incident ischemic stroke or first-ever stroke risk. This can blur the question the meta-analysis answers. Consider either (a) tightening inclusion to incident ischemic stroke risk in general populations, or (b) explicitly labeling strata (incident risk vs recurrence vs post-stroke outcomes) and avoiding cross-mixing in pooled estimates.

Inclusion/exclusion consistency (genetic risk vs stated criteria)

Methods state genetic risk factors were excluded to focus on modifiable, actionable factors (pp. 2–4), yet Table 1 lists a polygenic risk score paper (Marston et al., 2021). Please reconcile by either removing such studies from quantitative synthesis or explaining their role (e.g., descriptive only).

Heterogeneity and sensitivity analyses

I² is high for several factors (e.g., dyslipidemia 96%, smoking 94%; Table 2). The Methods say “No test was conducted to explore sources of heterogeneity” and “No sensitivity analyses were conducted”, but the Results later note leave-one-out sensitivity analyses were performed and were consistent. Please harmonize these statements and, if feasible, add simple sensitivity/subgroup checks (e.g., incident vs recurrent stroke; cohort vs cross-sectional; region; definition harmonization for smoking/obesity). Even a brief meta-regression with study-level covariates (design/region) could be informative.

Small-study effects testing

Egger’s tests are reported as non-significant for all risk factors. For several syntheses with k≤10 (e.g., AF, TIA), funnel-plot tests are underpowered and not recommended for firm inference. Suggest rephrasing to “underpowered to detect asymmetry; no clear evidence of small-study effects.”

Risk-of-bias tool application

You note using Newcastle–Ottawa Scale (NOS), but state that case–control studies are “not applicable” for NOS. NOS has a validated case–control version. Consider scoring all cohort and case–control studies and moving purely cross-sectional designs to narrative appraisal. This will strengthen the GRADE downgrading rationale.

Operational definitions and confounding

Key exposures vary across studies (e.g., smoking: current vs ever; obesity: BMI vs waist-to-hip ratio; dyslipidemia: composite vs LDL/HDL-specified). This likely contributes to the null for obesity/dyslipidemia. A short paragraph acknowledging definitional heterogeneity, statin use (for dyslipidemia), and mediation by co-existing metabolic syndrome would pre-empt criticism and align the discussion with Table 2 and Fig. 10.

Conclusions and antioxidant framing

The redox/neuroinflammation narrative is well-argued mechanistically, but the quantitative synthesis does not evaluate antioxidant interventions. Consider softening phrasing in the Abstract and Conclusions from “underline the potential benefits of antioxidant strategies” to “support the biological plausibility of exploring antioxidant strategies,” to avoid over-interpretation. See Abstract and Conclusions.

Minor comments

  • Decimal/thousand separators: Mixed use of commas/periods (e.g., OR = 1,575 vs 1.58; I² = 91,72). Standardize to journal style (likely periods for decimals). See Table 2.
  • OR vs RR labeling: Methods say pooled ORs; SoF table header says “OR/RR” . Clarify consistently.
  • Consistency of time window and access filters: Search limited to last 5 years and open-access English studies. Please acknowledge potential selection bias in the Limitations (you already note database scope; adding OA/English restriction would help).
  • PRISMA details: You mention reasons for exclusion are in Supplementary; consider adding a concise summary in the main text or PRISMA footnote (p. 5).
  • Formatting/track-changes artifacts: The Abbreviations and References pages show numerous “Formatted:” notes and repeated/garbled entries. A clean pass will greatly improve readability.
  • Abbreviations list: Deduplicate and align with text (e.g., repeated NO/ROS/NF-κB lines). Ensure all figure/table abbreviations are defined once. (pp. 22–23).
  • Title alignment: The title foregrounds “Proinflammatory Risk Factors,” yet AF and prior TIA are included (mechanistic links are thromboembolic/prognostic rather than inherently inflammatory). Consider “Modifiable Risk Factors and Redox-Inflammatory Pathways…” to match content.
  • Register the protocol (e.g., PROSPERO) and add the ID.
  • Provide a compact “definitions table” (exposure definitions used by each study), and a brief “direction-of-effect remains with statin-era confounding” note for dyslipidemia.
  • A one-paragraph clinical translation box: how to operationalize integrated risk-factor management (e.g., AF anticoagulation + BP and glycemic targets), consistent with your conclusions.

This is a timely synthesis that re-affirms the continued relevance of classic modifiable risk factors for ischemic stroke and thoughtfully weaves in a unifying redox/neuroinflammation framework. The Methods are broadly sound (REML random-effects; PRISMA-guided process), and the Summary of Findings table is very helpful for readers. The manuscript would benefit from a few clarifications in study selection, consistency of methods/results reporting, and presentation.

Major comments

Scope and study selection clarity

Table 1 includes heterogeneous study aims/populations (e.g., readmission after stroke, perioperative recurrent stroke after hip fracture, inflammatory indices in treated AIS, stenosis phenotypes), not strictly incident ischemic stroke or first-ever stroke risk. This can blur the question the meta-analysis answers. Consider either (a) tightening inclusion to incident ischemic stroke risk in general populations, or (b) explicitly labeling strata (incident risk vs recurrence vs post-stroke outcomes) and avoiding cross-mixing in pooled estimates.

Inclusion/exclusion consistency (genetic risk vs stated criteria)

Methods state genetic risk factors were excluded to focus on modifiable, actionable factors (pp. 2–4), yet Table 1 lists a polygenic risk score paper (Marston et al., 2021). Please reconcile by either removing such studies from quantitative synthesis or explaining their role (e.g., descriptive only).

Heterogeneity and sensitivity analyses

I² is high for several factors (e.g., dyslipidemia 96%, smoking 94%; Table 2). The Methods say “No test was conducted to explore sources of heterogeneity” and “No sensitivity analyses were conducted”, but the Results later note leave-one-out sensitivity analyses were performed and were consistent. Please harmonize these statements and, if feasible, add simple sensitivity/subgroup checks (e.g., incident vs recurrent stroke; cohort vs cross-sectional; region; definition harmonization for smoking/obesity). Even a brief meta-regression with study-level covariates (design/region) could be informative.

Small-study effects testing

Egger’s tests are reported as non-significant for all risk factors. For several syntheses with k≤10 (e.g., AF, TIA), funnel-plot tests are underpowered and not recommended for firm inference. Suggest rephrasing to “underpowered to detect asymmetry; no clear evidence of small-study effects.”

Risk-of-bias tool application

You note using Newcastle–Ottawa Scale (NOS), but state that case–control studies are “not applicable” for NOS. NOS has a validated case–control version. Consider scoring all cohort and case–control studies and moving purely cross-sectional designs to narrative appraisal. This will strengthen the GRADE downgrading rationale.

Operational definitions and confounding

Key exposures vary across studies (e.g., smoking: current vs ever; obesity: BMI vs waist-to-hip ratio; dyslipidemia: composite vs LDL/HDL-specified). This likely contributes to the null for obesity/dyslipidemia. A short paragraph acknowledging definitional heterogeneity, statin use (for dyslipidemia), and mediation by co-existing metabolic syndrome would pre-empt criticism and align the discussion with Table 2 and Fig. 10.

Conclusions and antioxidant framing

The redox/neuroinflammation narrative is well-argued mechanistically, but the quantitative synthesis does not evaluate antioxidant interventions. Consider softening phrasing in the Abstract and Conclusions from “underline the potential benefits of antioxidant strategies” to “support the biological plausibility of exploring antioxidant strategies,” to avoid over-interpretation. See Abstract and Conclusions.

Minor comments

  • Decimal/thousand separators: Mixed use of commas/periods (e.g., OR = 1,575 vs 1.58; I² = 91,72). Standardize to journal style (likely periods for decimals). See Table 2.
  • OR vs RR labeling: Methods say pooled ORs; SoF table header says “OR/RR” . Clarify consistently.
  • Consistency of time window and access filters: Search limited to last 5 years and open-access English studies. Please acknowledge potential selection bias in the Limitations (you already note database scope; adding OA/English restriction would help).
  • PRISMA details: You mention reasons for exclusion are in Supplementary; consider adding a concise summary in the main text or PRISMA footnote (p. 5).
  • Formatting/track-changes artifacts: The Abbreviations and References pages show numerous “Formatted:” notes and repeated/garbled entries. A clean pass will greatly improve readability.
  • Abbreviations list: Deduplicate and align with text (e.g., repeated NO/ROS/NF-κB lines). Ensure all figure/table abbreviations are defined once. (pp. 22–23).
  • Title alignment: The title foregrounds “Proinflammatory Risk Factors,” yet AF and prior TIA are included (mechanistic links are thromboembolic/prognostic rather than inherently inflammatory). Consider “Modifiable Risk Factors and Redox-Inflammatory Pathways…” to match content.
  • Register the protocol (e.g., PROSPERO) and add the ID.
  • Provide a compact “definitions table” (exposure definitions used by each study), and a brief “direction-of-effect remains with statin-era confounding” note for dyslipidemia.
  • A one-paragraph clinical translation box: how to operationalize integrated risk-factor management (e.g., AF anticoagulation + BP and glycemic targets), consistent with your conclusions.

Author Response

Comment 1: Table 1 includes heterogeneous study aims/populations (e.g., readmission after stroke, perioperative recurrent stroke after hip fracture, inflammatory indices in treated AIS, stenosis phenotypes), not strictly incident ischemic stroke or first-ever stroke risk. This can blur the question the meta-analysis answers. Consider either (a) tightening inclusion to incident ischemic stroke risk in general populations, or (b) explicitly labeling strata (incident risk vs recurrence vs post-stroke outcomes) and avoiding cross-mixing in pooled estimates.

Response 1: We thank the reviewer for highlighting this important point. Indeed, our dataset included studies with heterogeneous primary outcomes (incident ischemic stroke, recurrent stroke, and post-stroke outcomes). To address this, we have clarified the scope in the Methods section (lines 90-95), labeled the studies in Table 1 according to their primary outcome category, and acknowledged this as a limitation in the Discussion (lines 706-711). We retained all studies, as our rationale was that modifiable vascular risk factors exert consistent effects across the ischemic stroke continuum, but we agree that this may contribute to heterogeneity and reduce specificity.

Comment 2: Methods state genetic risk factors were excluded to focus on modifiable, actionable factors (pp. 2–4), yet Table 1 lists a polygenic risk score paper (Marston et al., 2021). Please reconcile by either removing such studies from quantitative synthesis or explaining their role (e.g., descriptive only).

Response 2: We agree and have reconciled the apparent inconsistency. The polygenic risk score paper (Marston et al., 2021) was not included in any pooled analysis (including DM2); it is now explicitly labeled ‘descriptive only’ in Table 1, and the Methods clarify that genetic studies were excluded from quantitative synthesis (lines 100-101).

Comment 3: I² is high for several factors (e.g., dyslipidemia 96%, smoking 94%; Table 2). The Methods say “No test was conducted to explore sources of heterogeneity” and “No sensitivity analyses were conducted”, but the Results later note leave-one-out sensitivity analyses were performed and were consistent. Please harmonize these statements and, if feasible, add simple sensitivity/subgroup checks (e.g., incident vs recurrent stroke; cohort vs cross-sectional; region; definition harmonization for smoking/obesity). Even a brief meta-regression with study-level covariates (design/region) could be informative.

Response 3: We thank the reviewer for this valuable observation. We have revised the Methods section to harmonize with the analyses actually performed. Specifically, we now state that, in addition to random-effects models (REML), we conducted leave-one-out sensitivity analyses for all risk factors (lines 160-166). For the four risk factors with the highest heterogeneity (hypertension, diabetes mellitus type 2, dyslipidemia, and smoking; I² > 90%), we also performed exploratory subgroup analyses by primary outcome (incident vs recurrent vs post-stroke), study design (cohort vs cross-sectional vs retrospective), and region when feasible. These additional results are reported in the Results section (lines 272-285, 287-300, 304-313, 316-330) and discussed in the Discussion, highlighting potential sources of variability and differences in effect estimates (lines 367-372, 381-392, 440-451). We further note that subgroup analyses for obesity and other factors were not feasible due to insufficient data per stratum. While we considered meta-regression, the limited number of studies per subgroup restricted the reliability of such analyses; we therefore prioritized reporting stratified analyses that are interpretable within the available evidence. Together, these revisions harmonize the Methods and Results, address sources of heterogeneity for the most heterogeneous factors, and strengthen the robustness of our findings.

Comment 4: Egger’s tests are reported as non-significant for all risk factors. For several syntheses with k≤10 (e.g., AF, TIA), funnel-plot tests are underpowered and not recommended for firm inference. Suggest rephrasing to “underpowered to detect asymmetry; no clear evidence of small-study effects.”

Response 4: We have revised both the Methods and Results sections to clarify the limitations of funnel plot and Egger’s test when the number of studies is small (k ≤ 10). The Methods now specify that such tests are underpowered and should be interpreted with caution (lines 171-173) . In the Results, the phrasing has been modified to: “Egger’s tests were non-significant across all risk factors. However, for syntheses including ≤10 studies (e.g., atrial fibrillation, TIA), the analyses were underpowered to reliably detect asymmetry; therefore, no clear evidence of small-study effects was observed.” (lines 266-268).

Comment 5: You note using Newcastle–Ottawa Scale (NOS), but state that case–control studies are “not applicable” for NOS. NOS has a validated case–control version. Consider scoring all cohort and case–control studies and moving purely cross-sectional designs to narrative appraisal. This will strengthen the GRADE downgrading rationale.

Response 5: We thank the reviewer for pointing this out. We have revised the risk-of-bias assessment by applying the appropriate Newcastle–Ottawa Scale checklists for both cohort and case–control studies. Purely cross-sectional studies were not formally scored, as NOS is not applicable, and were instead narratively appraised. Corresponding changes have been made in the Methods section (lines 175-178), in the Supplementary Table (NOS.xslx), and in the study limitations paragraph (lines 725-726). This also strengthens the rationale for GRADE downgrading, as suggested.

Comment 6: Key exposures vary across studies (e.g., smoking: current vs ever; obesity: BMI vs waist-to-hip ratio; dyslipidemia: composite vs LDL/HDL-specified). This likely contributes to the null for obesity/dyslipidemia. A short paragraph acknowledging definitional heterogeneity, statin use (for dyslipidemia), and mediation by co-existing metabolic syndrome would pre-empt criticism and align the discussion with Table 2 and Fig. 10.

Response 6: As suggested, we have added a paragraph in the Discussion (lines 356-366) explicitly acknowledging the heterogeneity in operational definitions of exposures across studies (e.g., smoking: current vs. ever, obesity: BMI vs. waist-to-hip ratio, dyslipidemia: composite vs. LDL/HDL fractions). We also noted the potential influence of statin use and the mediating role of metabolic syndrome, which may have contributed to the null associations observed for obesity and dyslipidemia. This addition aligns the interpretation of our results with Table 2 and Figure 10, and strengthens the clarity of our discussion.

Comment 7: The redox/neuroinflammation narrative is well-argued mechanistically, but the quantitative synthesis does not evaluate antioxidant interventions. Consider softening phrasing in the Abstract and Conclusions from “underline the potential benefits of antioxidant strategies” to “support the biological plausibility of exploring antioxidant strategies,” to avoid over-interpretation. See Abstract and Conclusions.

Response 7: We agree that our synthesis did not include interventional studies of antioxidant therapies, and therefore we have softened the phrasing in both the Abstract (lines 27-29) and Conclusions (lines 753-755). The revised wording now states that our findings “support the biological plausibility of exploring antioxidant strategies” rather than directly underlining their benefits. This adjustment prevents over-interpretation while maintaining alignment with the mechanistic rationale presented in the Discussion.

Comment 8: Decimal/thousand separators: Mixed use of commas/periods (e.g., OR = 1,575 vs 1.58; I² = 91,72). Standardize to journal style (likely periods for decimals). See Table 2.

Response 8: We thank the reviewer for pointing this out. We have standardized all decimal and thousand separators across tables and text to conform with journal style (periods for decimals, commas for thousands).

Comment 9: OR vs RR labeling: Methods say pooled ORs; SoF table header says “OR/RR” . Clarify consistently.

Response 9: We confirm that all pooled effect estimates in our meta-analysis were expressed as odds ratios (ORs). We have revised the Summary of Findings table header and all related text to consistently state “OR” rather than “OR/RR”.

Comment 10: Consistency of time window and access filters: Search limited to last 5 years and open-access English studies. Please acknowledge potential selection bias in the Limitations (you already note database scope; adding OA/English restriction would help).

Response 10: We have amended the Limitations section (lines 694-697) to explicitly acknowledge that our search was restricted to studies published in the last 5 years, in English, and in open-access format. We note that these criteria may have introduced selection bias, potentially omitting relevant studies outside this window or language scope.

Comment 11: PRISMA details: You mention reasons for exclusion are in Supplementary; consider adding a concise summary in the main text or PRISMA footnote (p. 5).

Response 11: We have added a concise summary of the main reasons for study exclusion in the Results section immediately above the PRISMA flow diagram (lines 197-200), while retaining the detailed list with references in the Supplementary Materials.

Comment 12: Formatting/track-changes artifacts: The Abbreviations and References pages show numerous “Formatted:” notes and repeated/garbled entries. A clean pass will greatly improve readability.

Response 12: We have removed all previous formatting artifacts from the Abbreviations and References sections. The revised file now shows only relevant tracked changes corresponding to our substantive edits, ensuring clarity and readability.

Comment 13: Abbreviations list: Deduplicate and align with text (e.g., repeated NO/ROS/NF-κB lines). Ensure all figure/table abbreviations are defined once. (pp. 22–23).

Response 13: We have revised the Abbreviations list by removing duplicate entries (e.g., NO, ROS, NF-κB), and ensured that each abbreviation is defined once and consistently. We also cross-checked all figure and table legends so that every abbreviation used in the manuscript is included in the list.

Comment 14: Title alignment: The title foregrounds “Proinflammatory Risk Factors,” yet AF and prior TIA are included (mechanistic links are thromboembolic/prognostic rather than inherently inflammatory). Consider “Modifiable Risk Factors and Redox-Inflammatory Pathways…” to match content.

Response 14: We respectfully maintain our current title. While atrial fibrillation and prior TIA are not purely inflammatory risk factors, our Discussion explicitly addresses their indirect contribution to redox-inflammatory pathways through thromboembolic mechanisms, endothelial dysfunction, and neuroinflammatory cascades following cerebral ischemia. As such, their inclusion under the broader concept of proinflammatory risk factors is consistent with the mechanistic framework presented in the manuscript.

Comment 15: Register the protocol (e.g., PROSPERO) and add the ID.

Response 15: We thank the reviewer for this suggestion. At the time of project initiation, the protocol was not registered in PROSPERO. Nevertheless, we strictly followed the PRISMA 2020 guidelines for systematic reviews, and the protocol (including objectives, eligibility criteria, and planned analyses) was predefined and is described in the Methods section. We also provide the completed PRISMA checklist as Supplementary Material. While we acknowledge that prospective registration is optimal, we believe the transparency of our reporting ensures reproducibility of the review process.

Comment 16: Provide a compact “definitions table” (exposure definitions used by each study), and a brief “direction-of-effect remains with statin-era confounding” note for dyslipidemia.

Response 16: We have added a table in Supplementary Material – “Definitions of dyslipidemia across studies.xlsx”, summarizing how dyslipidemia was defined across the included studies. We also revised Section 4.2 to note that, although the pooled estimate was not statistically significant, the direction-of-effect remained positive, but may have been attenuated by widespread statin use and treatment-related confounding (lines 453-460).

Comment 17: A one-paragraph clinical translation box: how to operationalize integrated risk-factor management (e.g., AF anticoagulation + BP and glycemic targets), consistent with your conclusions.

Response 17: In line with our conclusions, we have added a short “Clinical translation box” at the end of the Conclusions section (lines 762-769). This paragraph highlights how integrated risk-factor management can be operationalized in practice (e.g., combining anticoagulation for atrial fibrillation with evidence-based blood pressure, glycemic, and lipid targets, as well as smoking cessation). This addition strengthens the translational value of our work and aligns the conclusions with practical clinical implications.

Round 2

Reviewer 3 Report (New Reviewer)

The authors have satisfied my comments.

The authors have satisfied my comments.

This manuscript is a resubmission of an earlier submission. The following is a list of the peer review reports and author responses from that submission.

Round 1

Reviewer 1 Report

This work investigates the role of modifiable risk factors in ischemic stroke, with a particular focus on their contribution to neuroinflammation. The authors present a comprehensive and methodologically sound analysis, synthesizing data from 23 studies with over 690,000 participants. The systematic review and meta-analysis are well-executed and provide valuable insights into the relationship between stroke risk factors and the inflammatory pathways that drive cerebrovascular damage. Overall, the manuscript is well-organized, the study design is rigorous, and the conclusions are supported by strong evidence. Despite these strengths, there are some areas where more clarity or elaboration would help improve the manuscript's accessibility and rigor.

Specific Comments

Abstract, lines 17–24: The phrase "Emerging evidence suggests that oxidative stress mediates the impact of several modifiable risk factors..." could benefit from being more specific. It would be helpful to reference the key studies or biological mechanisms that support this claim. Including examples of the studies or mechanisms could help strengthen the argument and make the connection to ischemic stroke more direct.

Introduction, lines 42–46: The authors mention that stroke prevalence is higher in Eastern Europe, which is an important point. However, it would be valuable to explore how this higher prevalence impacts healthcare systems or stroke prevention strategies in these regions. Providing some context on how healthcare strategies should adapt or how targeted prevention efforts could be implemented would be useful.

Methods, lines 119–134: The inclusion and exclusion criteria are clearly stated, but the rationale for excluding genetic studies could be better explained. Given that genetic risk factors are an important part of stroke research, it would be beneficial to elaborate on why these studies were excluded, especially considering their potential relevance to the inflammatory pathways being explored.

Figure 3 legend: The legend for Figure 3 is informative, but additional details would improve its clarity. Specifically, it would be helpful to know whether the studies were grouped based on variables such as age, sex, or other demographic factors. Providing this information could enhance the interpretation of the pooled results, especially with regard to any potential subgroup differences.

Discussion, lines 312–328: The authors note that dyslipidemia did not show a significant association with ischemic stroke, which contrasts with previous studies. It would be helpful to discuss why this discrepancy may exist. Could variations in lipid profiles, patient populations, or study designs explain the lack of significance? A more detailed discussion on this would help clarify the reasons behind this finding.

Conclusion, lines 540–549: The authors mention the importance of personalized approaches in rehabilitation, but the connection between risk factors and rehabilitation strategies could be further explored. It would be valuable to expand on how the findings could influence personalized rehabilitation plans, particularly for patients with multiple overlapping risk factors. This would strengthen the conclusion and provide a more direct link between the study's findings and potential clinical applications.

Author Response

Thank you very much for taking the time to review this manuscript. Please find the detailed responses below and the corresponding revisions/corrections in track changes in the re-submitted files.

Comment 1: Abstract, lines 17–24: The phrase "Emerging evidence suggests that oxidative stress mediates the impact of several modifiable risk factors..." could benefit from being more specific. It would be helpful to reference the key studies or biological mechanisms that support this claim. Including examples of the studies or mechanisms could help strengthen the argument and make the connection to ischemic stroke more direct.

Response 1: We thank the reviewer for this helpful suggestion. We revised the abstract to provide more specificity, mentioning concrete biological mechanisms (lines 22-24).

Comment 2: Introduction, lines 42–46: The authors mention that stroke prevalence is higher in Eastern Europe, which is an important point. However, it would be valuable to explore how this higher prevalence impacts healthcare systems or stroke prevention strategies in these regions. Providing some context on how healthcare strategies should adapt or how targeted prevention efforts could be implemented would be useful.

Response 2: We added sentences to the introduction to emphasize the implications of the higher prevalence in Eastern Europe, highlighting the burden on healthcare systems and the need for targeted prevention strategies (lines 53-55).

Comment 3: Methods, lines 119–134: The inclusion and exclusion criteria are clearly stated, but the rationale for excluding genetic studies could be better explained. Given that genetic risk factors are an important part of stroke research, it would be beneficial to elaborate on why these studies were excluded, especially considering their potential relevance to the inflammatory pathways being explored.

Response 3: We appreciate this observation. We clarified that genetic studies were excluded because the aim of our work was to focus on modifiable, clinically actionable risk factors directly relevant for prevention strategies, whereas genetic predispositions, although important, cannot be modified through clinical or lifestyle interventions (lines 166-175).

Comment 4: Figure 3 legend: The legend for Figure 3 is informative, but additional details would improve its clarity. Specifically, it would be helpful to know whether the studies were grouped based on variables such as age, sex, or other demographic factors. Providing this information could enhance the interpretation of the pooled results, especially with regard to any potential subgroup differences.

Response 4: We clarified in the Results section that subgroup analyses by age or sex were not possible, as most included studies reported only aggregated data (lines 230-231).

Comment 5: Discussion, lines 312–328: The authors note that dyslipidemia did not show a significant association with ischemic stroke, which contrasts with previous studies. It would be helpful to discuss why this discrepancy may exist. Could variations in lipid profiles, patient populations, or study designs explain the lack of significance? A more detailed discussion on this would help clarify the reasons behind this finding.

Response 5: We thank the reviewer for pointing this out. We have expanded the discussion to address potential reasons for this discrepancy, including differences in lipid profile definitions (e.g., LDL vs HDL cholesterol), population characteristics, dietary patterns, and heterogeneity in study designs (lines 412-416).

Comment 6: Conclusion, lines 540–549: The authors mention the importance of personalized approaches in rehabilitation, but the connection between risk factors and rehabilitation strategies could be further explored. It would be valuable to expand on how the findings could influence personalized rehabilitation plans, particularly for patients with multiple overlapping risk factors. This would strengthen the conclusion and provide a more direct link between the study's findings and potential clinical applications.

Response 6: We agree with the reviewer. We have expanded the conclusion to better highlight the clinical implications of our findings for personalized rehabilitation, particularly in patients with overlapping risk factors (lines 685-693).

Reviewer 2 Report

Antioxidants

Title: Proinflammatory Risk Factors in Patients with Ischemic Stroke: A Systematic Review and Meta-Analysis

The study collects data and creates a meta-analysis about modifiable contributors to ischemic stroke, emphasizing their link to inflammatory pathways. The authors found significant associations between stroke risk and hypertension, smoking, diabetes and prior circulatory events.

This review and meta-analysis is potentially interesting, but the first part of the article, introduction and complete meta-analysis appears to be separate from the main review, since in the mate-analysis part of the article does not even mention any oxidative mechanisms, or anything related to the scope of the journal.

There are substantial needs for adjustment of the manuscript to make it consistent, coherent and well fit to the journal’s scope and make a considerable summary of the field.

The introduction is excessively long and digressive, including and start many subjects without deep diving into one main objective. After the introduction there is no substantial further information about the BBB, MMPs, Covid, NO, NF-kB, microvessel injuries, aging, etc. The authors should reduce the size of the introduction and make it more focused and consistent, also determining the main focus.

The manuscript starts to detail more about the connection of the oxidative stress and antioxidant system relatively late, and only minimally touches, detailing other mechanisms. This also reduces the consistency and questioning the manuscript fitting into the scope of the journal.

The manuscript only does the meta-analysis on a part of the otherwise known potential proinflammatory risk factors that are even partially listed in the introduction. This also fragments the review structure and leaves the reader questions about other factors missing from analysis, like aging, viral or bacterial infections, genetic disorders, etc.

The main subject of the review is indeed important and has relevance, but it is so deeply investigated and known that it is almost common sense in medicine and public health science. The main subject should be shifted towards the common factors like the oxidative stress and build the manuscript around that.

It would be highly recommended to prepare a summary figure which would show the weight of the different factors based on the meta-analysis.

It would be highly relevant to mention microbleeds which is novel recognition in hypertension that might affect brain homeostasis.

The last part of the review starting from “Overall, we identified several modifiable risk factors that were significantly” should start a separate paragraph like “Conclusions”.

Author Response

Thank you very much for taking the time to review this manuscript. Please find the detailed responses below and the corresponding revisions/corrections in track changes in the re-submitted files.

Comment 1: The introduction is excessively long and digressive, including and start many subjects without deep diving into one main objective. After the introduction there is no substantial further information about the BBB, MMPs, Covid, NO, NF-kB, microvessel injuries, aging, etc. The authors should reduce the size of the introduction and make it more focused and consistent, also determining the main focus.

Response 1: We appreciate the reviewer’s observation. Following the suggestion, we have substantially revised the Introduction and restructured it into four concise and coherent paragraphs. The first paragraph now focuses on the global and regional burden of ischemic stroke, highlighting its high prevalence in Eastern Europe and the implications for healthcare systems. The second paragraph summarizes the main etiological categories and core pathophysiological mechanisms, emphasizing the contribution of modifiable risk factors. The third paragraph introduces oxidative stress as a unifying biological pathway that links these risk factors to neuroinflammation and vascular injury, thereby aligning the Introduction more closely with the scope of the journal. Finally, the fourth paragraph presents recent epidemiological and lifestyle changes, underlines the limitations of previous meta-analyses, and clearly states the rationale and objectives of our systematic review and meta-analysis (pages 2,3).

Comment 2: The manuscript starts to detail more about the connection of the oxidative stress and antioxidant system relatively late, and only minimally touches, detailing other mechanisms. This also reduces the consistency and questioning the manuscript fitting into the scope of the journal.

Response 2: To improve consistency and to better align the manuscript with the scope of the journal, we have revised the Introduction to introduce oxidative stress earlier (lines 112-121), presenting it as a unifying pathway linking the main modifiable risk factors to neuroinflammation and vascular injury. In addition, we expanded the Discussion to more explicitly connect oxidative stress to the individual risk factors analyzed in our meta-analysis (hypertension, diabetes, dyslipidemia, smoking, alcohol consumption, and obesity) (lines 371-373, 420-421, 453-455, 510-511, 565-566, 578-595). Finally, the Conclusions were refined to emphasize the overarching role of oxidative stress as a common pathophysiological mechanism, thereby ensuring coherence between the mechanistic review and the meta-analytic findings (lines 676-684).

Comment 3: The manuscript only does the meta-analysis on a part of the otherwise known potential proinflammatory risk factors that are even partially listed in the introduction. This also fragments the review structure and leaves the reader questions about other factors missing from analysis, like aging, viral or bacterial infections, genetic disorders, etc.

Response 3: We thank the reviewer for this valuable observation. Our study was designed to specifically evaluate modifiable and clinically actionable risk factors for ischemic stroke. Therefore, we deliberately excluded factors such as aging, genetic predispositions, and infectious causes, as these are either non-modifiable or insufficiently standardized across the available literature to be directly translated into preventive strategies. To make this scope clear, we have now emphasized these criteria more explicitly in the Materials and Methods section, where we specify the rationale for excluding genetic and other non-modifiable contributors (lines 166-175). In addition, we have revised the Discussion (Limitations) section to acknowledge this decision (lines 635-639). This ensures that the focus of our review remains coherent with our main objective: identifying modifiable contributors with direct implications for prevention and rehabilitation strategies.

Comment 4: The main subject of the review is indeed important and has relevance, but it is so deeply investigated and known that it is almost common sense in medicine and public health science. The main subject should be shifted towards the common factors like the oxidative stress and build the manuscript around that.

Response 4: We appreciate the reviewer’s critical perspective. While it is true that modifiable risk factors for ischemic stroke are well recognized in clinical practice, the novelty of our study lies in providing an updated synthesis through systematic review and meta-analysis restricted to the last five years, contextualized within the rapidly changing lifestyle and epidemiological landscape. In line with the reviewer’s suggestion, we have reinforced throughout the Introduction, Discussion, and Conclusions sections the unifying role of oxidative stress as a common mechanistic pathway linking hypertension, diabetes, dyslipidemia, smoking, obesity, and alcohol consumption to cerebrovascular injury. This restructuring allows oxidative stress to serve as a central conceptual framework in our manuscript, while maintaining the clinical relevance of risk factor–specific analyses. We believe this integration strengthens both the consistency of the manuscript and its alignment with the journal’s scope.

Comment 5: It would be highly recommended to prepare a summary figure which would show the weight of the different factors based on the meta-analysis.

Response 5: In response, we have added a new summary figure (Figure 10) that graphically presents the pooled odds ratios with their 95% confidence intervals for each risk factor included in our meta-analysis. This figure complements the already existing Table 2, which provides the same data in tabular form, thus offering the reader both a numerical and a visual summary of the relative contribution of each factor (lines 298-301).

Comment 6: It would be highly relevant to mention microbleeds which is novel recognition in hypertension that might affect brain homeostasis.

Response 6: We agree that cerebral microbleeds represent an emerging and highly relevant aspect of hypertension-induced small vessel disease. Accordingly, we have now integrated this point into the Discussion section, highlighting microbleeds as metabolically active lesions reflective of chronic microvascular instability. We believe that this addition strengthens the pathophysiological link between hypertension, vascular remodeling, and ischemic injury (lines 383-389).

Comment 7: The last part of the review starting from “Overall, we identified several modifiable risk factors that were significantly” should start a separate paragraph like “Conclusions”.

Response 7: In accordance with the recommendation, we have revised the structure of the manuscript by modifying the initial “Overall…” paragraph from the Discussion and fully integrating its content into the newly designated Conclusions section. This adjustment enhances the clarity and logical flow of the manuscript, providing a more distinct separation between the interpretative discussion and the final take-home messages (lines 676-684).

Reviewer 3 Report

This is a relevant and well-structured review, but several key issues need to be addressed. The most significant limitation is the lack of a formal risk of bias assessment, which is essential in a systematic review. High heterogeneity (I² > 90% for several risk factors) is reported but not explored—subgroup or meta-regression analyses are needed to better interpret the findings. While GRADE is mentioned, its use is superficial and should be supported by a proper summary of findings table.

Additionally, there is a mismatch between the title and the methods. Although the paper emphasizes “proinflammatory” risk factors, the meta-analysis only includes traditional vascular risks without direct inflammatory markers. The mechanistic discussion on oxidative stress is informative but largely narrative and could be shortened or reframed to better fit the study design.

The introduction is lengthy and repeats well-known background material, which could be condensed. Figures (especially forest plots and PRISMA diagram) should be improved for clarity and resolution. Language is mostly fluent but would benefit from tighter editing to reduce redundancy. Finally, the disclosure of AI-assisted editing should be clarified in line with MDPI’s policy.

Author Response

Thank you very much for taking the time to review this manuscript. Please find the detailed responses below and the corresponding revisions/corrections in track changes in the re-submitted files.

Comment 1: The most significant limitation is the lack of a formal risk of bias assessment, which is essential in a systematic review.

Response 1: We thank the reviewer for this important comment. We have now performed a formal risk of bias assessment using the Newcastle–Ottawa Scale (NOS). Cohort and eligible cross-sectional studies were scored according to the NOS domains, while case–control studies and purely prevalence surveys were described narratively, as NOS criteria are not applicable for these designs. The results of this assessment are presented in Supplementary Materials (NOS.xlsx), and the Methods section has been updated accordingly (lines 234-238).

Comment 2: High heterogeneity (I² > 90% for several risk factors) is reported but not explored—subgroup or meta-regression analyses are needed to better interpret the findings.

Response 2: As described in the Results, we conducted leave-one-out sensitivity analyses, which showed that the overall findings remained consistent despite the high heterogeneity. We also qualitatively considered potential sources of heterogeneity, such as study design, geographic setting, and methodological quality. However, the number of studies available for each risk factor was too limited to allow reliable subgroup or meta-regression analyses without compromising statistical validity. We have clarified this point in the Methods and Limitations sections (lines 229-231,645-647).

Comment 3: While GRADE is mentioned, its use is superficial and should be supported by a proper summary of findings table.

Response 3: We thank the reviewer for this valuable comment. In the revised version of the manuscript, we have expanded the use of the GRADE approach beyond the brief mention in the initial submission. Specifically, we have now constructed a Summary of Findings table (Table 3), which presents for each risk factor the pooled effect estimate with 95% CI, the number of included studies and participants, the final certainty of evidence rating, and the reasons for downgrading. As all included studies were observational in design, the certainty of evidence started at Low according to GRADE recommendations and was further downgraded when inconsistency or imprecision were identified. The revised Methods (lines 239-246) and Results (lines 340-343) sections have been updated accordingly, and the Discussion (lines 617-625) now comments on the overall Low to Very Low certainty of evidence across the included risk factors.

Comment 4: Additionally, there is a mismatch between the title and the methods. Although the paper emphasizes “proinflammatory” risk factors, the meta-analysis only includes traditional vascular risks without direct inflammatory markers.

Response 4: We agree that our meta-analysis did not include direct inflammatory biomarkers, but rather focused on traditional vascular risk factors. The emphasis on “proinflammatory” in the title was intended to highlight the pathophysiological mechanisms through which these conventional factors (hypertension, diabetes, dyslipidemia, smoking, obesity, alcohol, atrial fibrillation, and prior TIA) exert their effects on ischemic stroke risk. To address this concern, we have revised the Introduction to explicitly state that the present analysis investigates frequently encountered, clinically relevant, and potentially modifiable vascular risk factors, which are known to act through inflammatory and oxidative pathways. In the Methods section, we have also clarified the rationale for including these risk factors and not extending the scope to other biomarkers or emerging proinflammatory markers (lines 166-175).

Comment 5: The mechanistic discussion on oxidative stress is informative but largely narrative and could be shortened or reframed to better fit the study design.

Response 5: In the revised manuscript, we have shortened the Introduction to avoid overly extensive mechanistic details and have revised the Discussion section to ensure that the comments on oxidative stress are concise and better aligned with the scope of our study.

Comment 6: The introduction is lengthy and repeats well-known background material, which could be condensed.

Response 6: We have condensed the Introduction by removing redundant background material and streamlining the text to provide a more concise and focused rationale for the study (pages 2,3).

Comment 7: Figures (especially forest plots and PRISMA diagram) should be improved for clarity and resolution.

Response 7: The figures included in the manuscript have been updated at high resolution and comply with the journal’s technical requirements. We have double-checked the clarity of labels and legends to ensure readability. Should the editorial office require specific formats or higher DPI, we will be happy to provide them.

Comment 8: Language is mostly fluent but would benefit from tighter editing to reduce redundancy.

Response 8: The manuscript has been carefully re-read and edited to improve fluency, reduce redundancy, and enhance overall clarity of the text.

Comment 9: Finally, the disclosure of AI-assisted editing should be clarified in line with MDPI’s policy.

Response 9: We thank the reviewer for this remark. In accordance with MDPI’s policy, we have clarified the disclosure statement. The revised manuscript explicitly states that AI-assisted editing was used for language refinement, under the supervision and full responsibility of the authors (lines 718-720).